# Muscle glycogen unavailability and fat oxidation rate during exercise: Insights from McArdle disease

Carlos Rodriguez-Lopez[1,2,3], Alfredo Santalla[4,5], Pedro. L Valenzuela[6], Alberto Real-Martínez[7,8], Mónica Villarreal-Salazar[7,8], Irene Rodriguez-Gomez[2,3] , Tomàs Pinós[7,8], Ignacio Ara[2,3] and Alejandro Lucia[3,6,9]

[1]*Department of Geriatrics, Hospital General Universitario Gregorio Marañón. Instituto de Investigación Sanitaria Gregorio Marañón (IiSGM), Madrid, Spain*

[2]*GENUD Toledo Research Group, Universidad de Castilla-La Mancha, Toledo, Spain*

[3]*CIBER of Frailty and Healthy Aging (CIBERFES), Madrid, Spain*

[4]*Department of Sport and Computer Science, Section of Physical Education and Sports, Faculty of Sport, Universidad Pablo de Olavide, Seville, Spain*

[5]*EVOPRED Research Group, Universidad Europea de Canarias, Tenerife, Spain*

[6]*Instituto de Investigación Sanitaria Hospital '12 de Octubre' ('imas12'), Madrid, Spain*

[7]*Mitochondrial and Neuromuscular Disorders Unit, Vall d'Hebron Institut de Recerca, Universitat Autònoma de Barcelona, Barcelona, Spain*

[8]*CIBER for rare disease (CIBERER), Madrid, Spain*

[9]*Faculty of Sport Sciences, Universidad Europea de Madrid, Madrid, Spain*

Handling Editors: Michael Hogan & Javier Gonzalez

The peer review history is available in the Supporting Information section of this article (https://doi.org/10.1113/JP283743#support-information-section).

C. Rodriguez-Lopez and A. Santalla contributed equally to this work.

T. Pinós, I. Ara, and A. Lucia share senior authorship.

**Abstract** Carbohydrate availability affects fat metabolism during exercise; however, the effects of complete muscle glycogen unavailability on maximal fat oxidation (MFO) rate remain unknown. Our purpose was to examine the MFO rate in patients with McArdle disease, comprising an inherited condition caused by complete blockade of muscle glycogen metabolism, compared to healthy controls. Nine patients (three women, aged $36 \pm 12$ years) and 12 healthy controls (four women, aged $40 \pm 13$ years) were studied. Several molecular markers of lipid transport/metabolism were also determined in skeletal muscle (gastrocnemius) and white adipose tissue of McArdle (*Pygm* p.50R*/p.50R*) and wild-type male mice. Peak oxygen uptake ($\dot{V}_{O_2peak}$), MFO rate, the exercise intensity eliciting MFO rate (FATmax) and the MFO rate-associated workload were determined by indirect calorimetry during an incremental cycle-ergometer test. Despite having a much lower $\dot{V}_{O_2peak}$ ($24.7 \pm 4$ *vs.* $42.5 \pm 11.4$ mL kg$^{-1}$ min$^{-1}$, respectively; $P < 0.0001$), patients showed considerably higher values for the MFO rate ($0.53 \pm 0.12$ *vs.* $0.33 \pm 0.10$ g min$^{-1}$, $P = 0.001$), and for the FATmax ($94.4 \pm 7.2$ *vs.* $41.3 \pm 9.1$ % of $\dot{V}_{O_2peak}$, $P < 0.0001$) and MFO rate-associated workload ($1.33 \pm 0.35$ *vs.* $0.81 \pm 0.54$ W kg$^{-1}$, $P = 0.020$) than controls. No between-group differences were found overall in molecular markers of lipid transport/metabolism in mice. In summary, patients with McArdle disease show an exceptionally high MFO rate, which they attained at near-maximal exercise capacity. Pending more mechanistic explanations, these findings support the influence of glycogen availability on MFO rate and suggest that these patients develop a unique fat oxidation capacity, possibly as an adaptation to compensate for the inherited blockade in glycogen metabolism, and point to MFO rate as a potential limiting factor of exercise tolerance in this disease.

(Received 17 August 2022; accepted after revision 31 October 2022; first published online 12 November 2022)

**Corresponding authors** I. Ara: GENUD Toledo Research Group, University of Castilla-La Mancha, Avda. Carlos III s/n, 45071, Toledo, Spain. Email: Ignacio.Ara@uclm.es; T. Pinós: Mitochondrial and Neuromuscular Disorders Unit, Vall d'Hebron Institut de Recerca (VHIR), Ps. Vall d'Hebron 119–129, 08035 Barcelona, Spain. Email: tomas.pinos@vhir.org

**Abstract figure legend** McArdle disease is caused by inherited blockade of glycogen breakdown in skeletal muscle fibres, with subsequent intolerance to most exercise tasks, as well as a substantial impairment of peak aerobic capacity. The present study indicates that the exercise capacity of these patients is mainly sustained by fat oxidation, with active patients showing an exceptional maximal fat oxidation rate (comparable to athletes) during endurance exercise, possibly as an adaptation to muscle glycogen unavailability. On the other hand, data in the (untrained) mouse model of the disease revealed overall no major differences at baseline in molecular markers of lipid transport/metabolism compared to wild-type mice.

## Key points

- Physically active McArdle patients show an exceptional fat oxidation capacity.
- Maximal fat oxidation rate occurs near-maximal exercise capacity in these patients.
- McArdle patients' exercise tolerance might rely on maximal fat oxidation rate capacity.
- Hyperpnoea might cloud substrate oxidation measurements in some patients.
- An animal model revealed overall no higher molecular markers of lipid transport/metabolism.

**Carlos Rodriguez-Lopez** completed his PhD in public health and physical activity research at the University of Castilla-La Mancha (Toledo, Spain). During this period, he had the opportunity to work with patients with McArdle disease and began to become interested in learning more about the physiological responses and adaptations to exertion of these individuals. **Alfredo Santalla**, PhD, is an Associate Professor in Exercise Physiology at the Universidad Pablo Olavide (Seville, Spain). He has extensively published in McArdle disease, being the first author of the last update of the registry of Spanish patients with this condition. He altruistically assists patients with this condition from all over the world, with personalized advice on nutrition and exercise. One of his novel contributions to the management of patients is his finding on the safety and efficacy of supervised resistance training, which is now recommended in the international clinical guidelines for the management of McArdle disease.

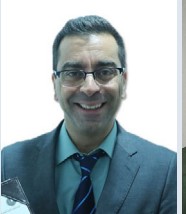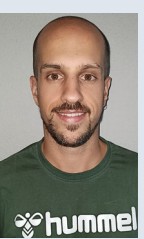

## Introduction

McArdle disease (glycogenosis type V) is an autosomal recessive disease caused by pathogenic mutations in the gene (*PYGM*) encoding the skeletal muscle isoform of glycogen phosphorylase, also known as 'myophosphorylase' (Lucia et al., 2008; Santalla et al., 2014). Myophosphorylase catalyses the sequential phosphorolysis of glycogen by removing (1,4)-$\alpha$-glucosyl units from the outer branches to release glucose 1-phosphate. As such, muscle glycogen metabolism is blocked in affected patients and ATP generation during endurance exercise relies on the oxidation of circulating glucose and fatty acids. A unique hallmark of McArdle disease is that, during the start of an endurance exercise task (i.e. when blood substrate availability for muscle fibres is still limited) involving large muscle groups (e.g. brisk walking, swimming, bicycling), patients experience premature fatigue, tachycardia and myalgia and are at risk of contractures. These indicators of early exercise intolerance are, however, attenuated (i.e. the so-called 'second wind' phenomenon) after 6–10 min, coinciding with an increased availability of blood substrates (glucose and fatty acids) (Salazar-Martínez et al., 2021; Vissing & Haller, 2003a).

A better understanding of fat oxidation in patients with McArdle disease is of interest because this disorder represents a unique model in exercise physiology and muscle metabolism (Kitaoka, 2014; Santalla et al., 2014). Notably, it allows the investigation of how total muscle glycogen unavailability might affect muscle responses/adaptations to exercise without the need for dietary (e.g. extreme restrictions in carbohydrate intake) or pharmacological manipulation (Fiuza-Luces et al., 2018). For example, in a preclinical study using a 'knockin' mouse model of McArdle disease, which harbours the most common pathogenic *Pygm* gene mutation in Caucasians (i.e. p.R50*) and closely mimics the main disease manifestations in patients (Nogales-Gadea et al., 2012), we found remarkable differences in the protein networks involved in muscle tissue adaptations induced by endurance exercise training compared to wild-type (WT) mice (Fiuza-Luces et al., 2018). Other studies examining the skeletal muscle of patients with McArdle disease have reported an enhanced extra-muscular fuel uptake (i.e. monocarboxylate transporters) (Kitaoka, Ogborn, Mocellin et al., 2013) and an upregulation of the myo-adenylate deaminase–xanthine oxidase pathway (Kitaoka, Ogborn, Nilsson et al., 2013), which might represent cellular adaptations to compensate for the blocked glycogenolysis. However, few studies have specifically assessed the rate of fat oxidation during exertion (e.g. as determined with indirect calorimetry) in patients with McArdle disease (Ørngreen et al., 2009; Ørngreen et al., 2015). Moreover, to the best of our knowledge, no study

has determined the fat oxidation capacity of these patients at different exercise intensities. This information might be used to gain insight into muscle metabolic responses to exertion in the event of muscle glycogen unavailability.

The main aim of the present study was to examine the fat oxidation rate of patients with McArdle disease at different submaximal exercise intensities in comparison with sex- and age-matched healthy controls. We hypothesized that physically active patients will exhibit a higher maximal fat oxidation (MFO) rate (g min$^{-1}$) than controls. We complemented our findings with studies in McArdle mice, by examining the levels of several proteins involved in fat transport/oxidation in skeletal muscle and white adipose tissue (WAT).

## Methods

### Study in patients

**Ethical approval.** This case–control study was approved by the ethics committee of the Research Institute of the Hospital 12 de Octubre (Madrid, Spain; reference #16/081) and adhered to the tenets of the *Declaration of Helsinki*, except for registration in a database. Participants were informed of the aims and procedures of the study, as well as the possible risks and benefits, giving their written consent. Assessment took place during 2019 (patients) and 2021 (controls) at the Faculty of Sports Sciences of Universidad Castilla-La Mancha (Toledo, Spain).

**Participants.** Individuals diagnosed with McArdle disease were recruited through the Spanish National Registry of patients with this condition (Santalla et al., 2017). The presence of McArdle disease was confirmed by genetic diagnosis in one of the three 'reference' centres for genetic analysis (Hospital 12 de Octubre, Madrid; Hospital Val d'Hebron, Barcelona; or Hospital Meixoeiro, Vigo). This implied identification of a documented pathogenic mutation causing McArdle disease in both *PYGM* alleles (either the same mutation in homozygosis or two different mutations in heterozygosis), as per international recommendations (Lucia et al. 2021). Mutant *PYGM* alleles were identified in muscle or blood samples using SNaPShot minisequencing (Thermo Fisher, Waltham, MA, USA) (Rubio et al., 2007), followed by Sanger sequencing of the entire coding region and intron/exon boundaries (Kubisch et al., 1998). Alternatively, a next-generation sequencing-customized gene panel on a PGM-IonTorrent platform (Thermo Fisher), consisting of 35 genes (including *PYGM*) associated with metabolic myopathies, was used. An age- and sex-matched control group of healthy, non-athletic subjects was recruited through local advertisements.

The following common inclusion criteria were established for the two groups: age 16–65 years, having no condition (notably, pregnancy) contra-indicating dual-energy X-ray absorptiometry (DXA), being free of any major cardiorespiratory disease or severe condition contraindicating exercise and not being enrolled in structured exercise training or sports competition. Patients had to be physically active (i.e. $\geq$150 min week$^{-1}$ of moderate-intensity aerobic activities or $\geq$75 min week$^{-1}$ of vigorous-intensity aerobic activities, or a combination thereof) and belong to the two lowest classes (0 or 1) of clinical severity scale for McArdle disease: the 'Martinuzzi scale' [ranging from 0 (lowest) to 3 (highest)] (Martinuzzi et al., 2003); where 0 = asymptomatic or virtually asymptomatic (mild exercise intolerance, but no functional limitation in any daily life activity); 1 = exercise intolerance, contractures, myalgia, and limitation of acute strenuous exercise, and occasionally in daily life activities; no record of myoglobinuria, no muscle wasting or weakness; 2 = same as 1, plus recurrent exertional myoglobinuria, moderate restriction in exercise and limitation in daily life activities; and 3 = same as 2, plus fixed muscle weakness, with or without wasting and severe limitations on exercise and most daily life activities. Finally, patients also had to be familiarized with exercise testing in our laboratory. This included having carried out one or more 12 min sessions of cycle-ergometer exercise testing at a constant submaximal workload to pass the second wind (Salazar-Martínez et al., 2021) and having previously experienced the feeling of myalgia associated with heavy exertion (i.e. having reached at least once a level of rating of perceived pain (RPP) (Jensen & McFarland, 1993) of 9–10 on a scale from 0 to 10).

**Assessments.** All participants attended the laboratory between 07.00 h and 10.00 h after an overnight fast of 12–14 h in duration. They were asked to refrain from strenuous physical activity and to avoid alcohol, tobacco and caffeine consumption the day before assessments. Before exercise testing, basic anthropometric data (body mass, height) were obtained with a precision scale-stadiometer (model 711; Seca, Hamburg, Germany), followed by body composition assessment (whole-body fat and fat-free mass) using a calibrated DXA device (QDR Discovery Wi; Hologic, Bedford, MA, USA). Details on the DXA measurements are reported elsewhere (Rodríguez-Gómez et al., 2020).

Thereafter, participants completed an exercise test protocol as detailed below. All exercise tests were performed using the same cycle-ergometer (model 800S; Ergoline, Bitz, Germany) and metabolic cart for indirect calorimetry measurements (Oxycon Pro; Jaeger, Hoechberg, Germany), with the latter synchronized with

a heart rate monitor (model A300; Polar electro Oy, Kempele, Finland). Ventilatory (minute ventilation [VE] and breathing rate) and gas exchange data [oxygen uptake ($\dot{V}_{O_2}$), carbon dioxide production ($\dot{V}_{CO_2}$), respiratory exchange ratio (RER), end-tidal pressure of carbon dioxide ($P_{ETCO_2}$) and ventilatory equivalent of oxygen (VE·$\dot{V}_{O_2}^{-1}$) and carbon dioxide (VE·$\dot{V}_{CO_2}^{-1}$)] were recorded 'breath-by-breath' during exercise.

Before data collection, patients performed a 12 min exercise bout at constant moderate intensity to pass the second wind, as described previously (Salazar-Martínez et al., 2021). Thereafter, they rested in a seated position for 5–10 min before starting the incremental exercise test in order to eliminate residual fatigue.

To determine MFO rate during exercise, all participants completed an incremental protocol at a pedaling cadence of 60–90 r.p.m. After 3 min during which time they remained seated in the ergometer (i.e. 0 W) to stabilize gas-exchange data, the workload was increased by ~15 (women) or ~20 W (men) every 3 min until RER $\geq$1.00 (controls) or until exhaustion (patients, see below). Workload increments were slightly adapted in some participants to ensure that a minimum of four 3 min stages were completed by those with a lower fitness level, and to avoid excessive test duration (i.e. >30 min) in the fittest participants. At the end of each 3 min stage, we determined the RPP and the rating of perceived exertion (RPE, on a scale from 0 to 10 ) in McArcdle patients (Salazar-Martínez et al., 2021). In this group, 'exhaustion' was determined when one of the following criteria was met: start (or near-start) of muscle contracture(s), unable to maintain the required workload (pedal cadence <60 r.p.m.) and RPE or RPP >9. The $\dot{V}_{O_2peak}$ and peak work capacity of the patients were determined as the highest $\dot{V}_{O_2}$ value obtained during a 20 s average and the corresponding workload, respectively.

Following 5 min of recovery after the incremental exercise protocol, participants in the control group completed a maximal ramp protocol for $\dot{V}_{O_2peak}$ (highest 20 s average) and peak work capacity determination. The test started with a 3 min stage at the same wattage as in the last stage of the previous submaximal incremental protocol and, thereafter, the workload was increased by 3 W (women) or 4 W (men) every 12 s until volitional exhaustion. The $\dot{V}_{O_2peak}$ and peak work capacity were determined as explained above for patients.

Substrate oxidation was determined in all participants by means of gas exchange measurements for each 3 min stage of the incremental protocol. Fat oxidation rate (g min$^{-1}$) was calculated according to Frayn's stoichiometric equations with the assumption that urinary nitrogen excretion was 0 g (Frayn, 1983) at the same time as considering the average of $\dot{V}_{O_2}$ and $\dot{V}_{CO_2}$ values during the last 60 s of each 3 min stage (Amaro-Gahete, Sanchez-Delgado, Alcantara et al., 2019).

Individuals' relationships between fat oxidation rate and relative exercise intensity (%$\dot{V}_{O_2peak}$) were determined using a second-order polynomial curve. MFO rate, exercise intensity eliciting MFO rate (also known as 'FATmax') and the workload eliciting the MFO rate were identified through interpolation. Each polynomial curve was visually inspected by an experimented evaluator to ensure the best goodness-of-fit, discarding those stages or values potentially biased by aberrant ventilatory patterns or artifacts. Additionally, to compare fat oxidation rates at the same absolute workload (0, 20, 40, 60, 80 and 100 W) and relative intensity (30%, 40%, 50% and 60% of $\dot{V}_{O_2peak}$) between groups, interpolated values were calculated from the relationship between fat oxidation rate and workload and between fat oxidation rate and relative exercise intensity, respectively. Workload and $\dot{V}_{O_2peak}$ values were normalized to body mass, whereas fat oxidation rate was computed in absolute values and relative to fat-free mass to reduce the influence of body composition.

## Study in mice

All experimental procedures were approved by the Vall d'Hebron Institutional Review Board (protocol number 58/17 CEEA; 35/04/08) and conducted in accordance with the European Convention for the Protection of Vertebrate Animals used for Experimental and Other Scientific Purposes (ETS1 2 3) and Spanish laws (32/2007 and R.D. 1201/2005). Previously developed McArdle (p.R50*/p.R50*) male mice, back-crossed for 10 generations onto the C57/6J background (Nogales-Gadea et al., 2012), were used in the study. For molecular analyses, seven control (WT, p.R50R/p.R50R) and seven McArdle male mice aged between 8 and 20 weeks were used. Mice were sacrificed by cervical dislocation immediately before dissection of hind-limb muscles (gastrocnemius) and WAT. The gastro-cnemius muscle was chosen for the same reason as in previous research by our group with the McArdle mouse (Brull et al., 2015), in that this is overall an 'inter-mediate' skeletal muscle in terms of the predominant metabolic phenotype as opposed to other muscles with a more characteristic oxidative/slow-twitch (e.g. soleus) or glycolytic/fast-twitch (e.g. extensor digitorum longus) profile, thereby providing a representation of the biological responses in the different main fibre types (I, IIA and IIB) that can be found in the adult mouse skeletal muscle.

**Western blotting.** Gastrocnemius muscle and WAT samples, were homogenized using a Pellet Pestle Cordless motor (Sigma-Aldrich, Madrid, Spain) in cold homo-genization buffer (Tris-HCl, 20 mM; NaCl, 150 mM; and Triton X-100, 1%) and centrifuged (10 000 ***g***)

for 10 min at 4°C. The samples were then placed in boiling water for 3 min, and 30 $\mu$g of total protein was applied to each lane of an SDS-PAGE gel for electro-phoresis followed by membrane transfer. Non-specific binding of membranes was blocked by incubation in 5% low-fat dried milk powder in phosphate-buffered saline. Membranes were incubated with primary antibodies over-night at 4°C (Table 1) and subsequently with secondary antibodies for 3 h at room temperature. Ponceau S staining solution (Sigma-Aldrich) was used as a loading control for all analyses. Membranes were developed with the Immobilon Western Chemiluminiscent HRP Substrate (Merck-Millipore, Burlington, MA, USA) and images were obtained with a LICOR Odyssey® Fc Imaging System (LICOR Biosciences, Lincoln, NE, USA) and quantified with Image Studio™ Lite software, version 5.2 (LICOR Biosciences). We determined the levels of the following proteins involved in lipid transport and/or metabolism in skeletal muscle or WAT:

- 3-Hydroxyacyl-CoA dehydrogenase (HADH), a mitochondrial fatty acid $\beta$-oxidation enzyme that catalyses the third step of $\beta$-oxidation.
- Adipose triglyceride lipase (ATGL) phosphorylated at serine 406 (pATGL Ser 406), an enzyme that catalyses the first reaction of lipolysis, where triacylglycerols are hydrolysed to diacylglycerols.
- Total and phosphorylated hormone-sensitive lipase (HSL, pHSL Ser565 and pHSL Ser660, respectively), one of the three lipases that is especially important during periods of energy demand and is tightly controlled by insulin (Lan et al., 2019).
- AMP-activated protein kinase [AMPK, total and phosphorylated levels (pAMPK Thr172)]. AMPK is a key regulator of skeletal muscle fat metabolism, mainly by regulating fatty acid transport into the mitochondria via interactions with acetyl-CoA carboxylase, malonyl-CoA decarboxylase, or the fatty acid transporter transmembrane glycoprotein cluster of differentiation 36 (CD36), or by phosphorylation of transcription factors such as peroxisome proliferator-activated receptor gamma coactivator 1 alpha (PGC-1$\alpha$) (Thomson & Winder, 2009).
- CD36, a scavenger receptor class B protein that serves various functions in lipid metabolism and signalling, particularly in facilitating the cellular uptake of long-chain fatty acids (Glatz et al., 2022).
- Perilipin 5 (Plin5), a member of the perilipin super-family that is a lipid droplet-targeting protein central to lipid homeostasis in highly oxidative tissues like skeletal muscle, where it promotes association of lipid droplets with mitochondria (Zhang et al., 2022) and prevents excessive fat accumulation (Kimmel & Sztalryd, 2014).

**Table 1. List of antibodies used for western blotting**

| | Primary antibody | | | | Secondary antibody | | |
|---|---|---|---|---|---|---|---|
| ID | Reference | Dilution | Diluent | ID | Dilution | Diluent |
| CD36 | GeneTex/ GTX100642 | 1:2000 | TTBS 0.1% + 5% Skim milk | Goat to Rabbit HRP (DAKO P0448) | 1:5000 | TTBS 0.1% + 5% Skim milk |
| pATGLSer406 | Abcam/ab135093 | 1:750 | TTBS 0.1% + 5% Skim milk | Goat to Rabbit HRP (DAKO P0448) | 1:5000 | TTBS 0.1% + 5% Skim milk |
| HSL Total | Cell Signalling/ 4107 | 1:2000 | TTBS 0.1% + 5% Skim milk | Goat to Rabbit HRP (DAKO P0448) | 1:5000 | TTBS 0.1% + 5% Skim milk |
| pHSLSer565 | Cell Signalling/ 4137 | 1:1000 | TTBS 0.1% + 5% Skim milk | Goat to Rabbit HRP (DAKO P0448) | 1:5000 | TTBS 0.1% + 5% Skim milk |
| pHSLSer660 | Cell Signalling/ 4126 | 1:1000 | TTBS 0.1% + 5% Skim milk | Goat to Rabbit HRP (DAKO P0448) | 1:5000 | TTBS 0.1% + 5% Skim milk |
| Plin 5 | Thermo Scientific/ PA1-46 215 | 1:15 000 | TTBS 0.1% + 5% Skim milk | Goat to Rabbit HRP (DAKO P0448) | 1:5000 | TTBS 0.1% + 5% Skim milk |
| HADH | Thermo Scientific/ PA5-31 157 | 1:2000 | TTBS 0.1% + 5% Skim milk | Goat to Rabbit HRP (DAKO P0448) | 1:5000 | TTBS 0.1% + 5% Skim milk |
| AMPK | Cell Signalling/ 2532 | 1:2000 | TTBS 0.1% + 5% Skim milk | Goat to Rabbit HRP (DAKO P0448) | 1:5000 | TTBS 0.1% + 5% Skim milk |
| pAMPKThr172 | Cell Signalling/ 2531 | 1:2000 | TTBS 0.1% + 5% Skim milk | Goat to Rabbit HRP (DAKO P0448) | 1:5000 | TTBS 0.1% + 5% Skim milk |

Abbreviations: AMPK, AMP-activated protein kinase; CD36, cluster of differentiation 36; HADH, 3-hydroxyacyl-CoA dehydrogenase; FBS, foetal bovine serum; HSL, hormone-sensitive lipase; pAMPKThr172, AMP-activated protein kinase phosphorylated at threonine 172; pATGL Ser 406, adipose triglyceride lipase (ATGL) phosphorylated at serine 406; PBS, phosphate-buffered saline; pHSL Ser565, hormone-sensitive lipase (HSL) phosphorylated at serine 565; pHSL Ser660, hormone-sensitive lipase (HSL) phosphorylated at serine 660; Plin5, perilipin 5; TTBS, Tris-buffered saline + Tween 20;WB, western blotting;

## Statistical analysis

Descriptive data are presented as the mean ± SD unless otherwise stated. The distribution of the data was investigated with the Shapiro–Wilk test. For studies with patients and controls, between-group differences were determined by Student's *t* test for independent samples. In addition, because of the potential influence of cardiorespiratory fitness ($\dot{V}_{O_2peak}$) on MFO (Amaro-Gahete, Sanchez-Delgado, Ara et al., 2019), an analysis of covariance was conducted to compare MFO rate between groups including $\dot{V}_{O_2peak}$ as a covariate. Effect sizes (Hedges' *g*) were computed to analyse the magnitude of the differences (Lakens, 2012) and these were described as: small (>0.2), moderate (>0.5) or large (>0.8) (Hopkins et al., 2009). The Mann–Whitney *U* test was used for comparisons between McArdle and WT mice. Prism, version 8.0 (GraphPad Software Inc., San Diego, CA, USA) was used for the analyses. $P < 0.05$ was considered statistically significant.

## Results

### Study in patients

Nineteen patients with McArdle disease (eight women) and 12 healthy controls (four women) who met all inclusion criteria and provided informed consent were evaluated (participants' flowchart shown in Fig. 1). The different *PYGM* genotypes of the 19 patients with McArdle disease are shown in Table 2. All pathogenic *PYGM* mutations identified in the present study have been previously reported in the last update of the Spanish registry of patients with McArdle disease (Santalla et al., 2017). Furthermore, all reported *PYGM* genotypes result in complete deficiency of myophosphorylase activity and thus in a complete inability to use muscle glycogen as a metabolic substrate (García-Consuegra et al., 2018; Nogales-Gadea et al., 2008; Nogales-Gadea et al., 2015).

Overnight fasting lasted 13 ± 2 and 13 ± 1 h in patients and controls, respectively ($P = 0.767$ for the between-group difference). Nine patients were excluded from the final analyses because their gas-exchange data reflected abnormalities (as a result of excessive hyperpnoea) that precluded the correct assessment of fat oxidation, and another patient failed to complete four 3 min stages of the exercise test. Accordingly, nine patients (three women) were finally included in the comparative analyses. Regarding the aforementioned nine patients excluded from the study analyses as a result of excessive hyperpnoea, aiming to examine a potential 'confounding' influence of this phenomenon on fat oxidation rate, we compared the mean values of both $P_{ETCO_2}$ and $VE \cdot \dot{V}_{CO_2}^{-1}$

at the last 3 min stage of the incremental exercise test used for fat oxidation determination (normalized to the values registered in the first stage of this test) between these patients, the rest of patients and the control group (Fig. 2). The results (one-way analysis of variance) (Fig. 2) showed significantly lower and higher values of 'normalized' $P_{ETCO_2}$ and $VE \cdot \dot{V}_{CO_2}^{-1}$, respectively, in the nine 'excluded' patients compared to both the rest of patients and the control group, denoting marked hyperpnoea in the former.

The main characteristics of the participants that were finally included in the study analyses are shown in Table 3. No between-group differences were found for sex, age, or body composition, but both $\dot{V}_{O_2peak}$ and peak work capacity were significantly lower (both $P < 0.0001$, with large effect sizes) in patients compared to controls.

Fat oxidation rates at increasing intensities in the two groups are shown in Fig. 3 [individual values in Fig. 3*A* (patients) and Fig. 3*B* (controls)]. At low relative exercise intensity (30% of $\dot{V}_{O_2peak}$) patients showed

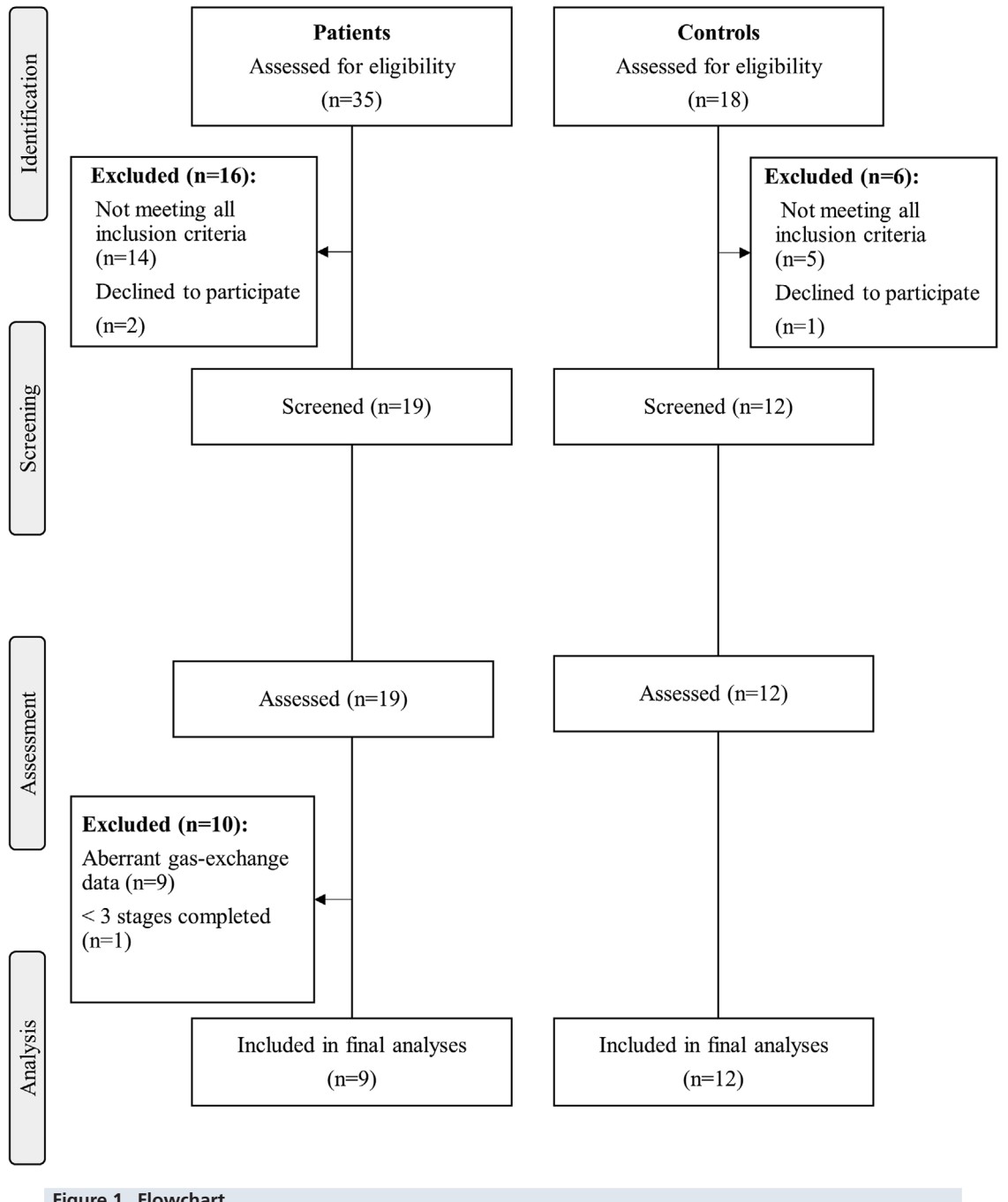

**Figure 1. Flowchart**
Participant flowchart.

**Table 2. Pathogenic *PYGM* genotype indicative of McArdle disease identified in all studied patients (*n* = 19)**

| Patient | Mutation in each *PYGM* allele | Disease severity class* | FATmax data included in analyses? |
|---|---|---|---|
| 1 | p.R50* (c.148C>T) / p.R50* (c.148C>T) | 1 | Yes |
| 2 | p.R50* (c.148C>T) / p.R50* (c.148C>T) | 1 | No |
| 3 | p.R50* (c.148C>T) / p.R50* (c.148C>T) | 1 | No |
| 4 | p.R50* (c.148C>T) / p.R50* (c.148C>T) | 0 | Yes |
| 5 | p.R50* (c.148C>T) / p.R50* (c.148C>T) | 1 | No |
| 6 | p.R50* (c.148C>T) / p.W798R (c.2392 T>C) | 1 | Yes |
| 7 | p.R50* (c.148C>T) / p.W798R (c.2392 T>C) | 1 | Yes |
| 8 | p.R50* (c.148C>T) / p.W798R (c.2392 T>C) | 0 | No |
| 9 | p.G205S (c.613G>A) / p.G205S (c.613G>A) | 1 | Yes |
| 10 | p.G205S (c.613G>A) / p.G205S (c.613G>A) | 1 | No |
| 11 | p.G205S (c.613G>A) / p.G205S (c.613G>A) | 1 | No |
| 12 | p.R50X (c.148C>T) / p.T488 N (c.1463C>A) + p.K215 K (c.645G>A) | 1 | Yes |
| 13 | p.R50X (c.148C>T) / p.T488 N (c.1463C>A) + p.K215 K (c.645G>A) | 1 | No |
| 14 | p.R50* (c.148C>T) / p.K754fs*49 (c.2262delA) | 1 | Yes |
| 15 | p.R50* (c.148C>T) / p.R602W (c.1804C>T) | 1 | No |
| 16 | p.R50* (c.148C>T) / p.A660D (c.1979C>A) | 1 | Yes |
| 17 | p.R50* (c.148C>T) / p.E383K (c.1147G>A) | 1 | No |
| 18 | p.G205S (c.613G>A) / c.1768 + 1G>A | 1 | No |
| 19 | p.R50* (c.148C>T) / p.A55Gfs*21 (c.163_167delGCTCT) | 1 | Yes |

Abbreviations: FAT$_{max}$, exercise intensity eliciting the maximum fat oxidation rate. Symbol: *Assessed with the so-called 'Martinuzzi scale' [ranging from 0 (lowest) to 3 (highest)] (Martinuzzi et al., 2003), where: 0 = asymptomatic or virtually asymptomatic (mild exercise intolerance, but no functional limitation in any daily life activity); 1 = exercise intolerance, contractures, myalgia, and limitation of acute strenuous exercise, and occasionally in daily life activities; no record of myoglobinuria, no muscle wasting or weakness; 2 = same as 1, plus recurrent exertional myoglobinuria, moderate restriction in exercise, and limitation in daily life activities; 3 = same as 2, plus fixed muscle weakness, with or without wasting and severe limitations on exercise and most daily life activities.

lower fat oxidation rates than controls ($P = 0.010$, $g = 1.74$), whereas no differences were found at higher (moderate) intensities (i.e. 40%, 50% and 60% of $\dot{V}_{O_2peak}$; $P = 0.095$, 0.669 and 0.106, respectively) (Fig. 3*C*). No between-group differences were found for fat oxidation rates at 0, 20, or 40 W ($P = 0.149$, $P = 0.871$ and $P = 0.141$, respectively); however, patients showed significantly higher fat oxidation rates (with large effect sizes) at higher workloads [$P = 0.026$ ($g = 1.08$) for 60 W; $P = 0.005$ ($g = 1.48$) for 80 W; and $P < 0.0001$ ($g = 2.00$) for 100 W] (Fig. 3*D*). Overall, the goodness-of-fit ($R^2$) of the relationship between individual values of fat oxidation rates and workload or relative exercise intensity was excellent ($0.950 \pm 0.041$).

Patients with McArdle disease showed significantly higher values for the MFO rate (with large effect sizes) than controls regardless of whether this variable was expressed in 'absolute' units (g min$^{-1}$; $P = 0.001$, $g = 1.79$) or relative to fat-free mass (mg kg$^{-1}$ min$^{-1}$; $P < 0.0001$, $g = 2.11$) (Fig. 4*A*). The exercise intensity eliciting the MFO rate (i.e. FATmax) was also higher in patients whether expressed as %$\dot{V}_{O_2peak}$ ($P < 0.0001$, $g = 6.47$) (Fig. 4*B*) or in W kg$^{-1}$ ($P = 0.020$, $g = 1.16$) (Fig. 4*C*).

### Study in mice

No differences were found in the levels of CD36, HSL, pHSLSer565, pHSLSer565/HSL ratio or Plin5 in muscle between WT and McArdle mice, indicating no overall differences in lipid mobilization or utilization in this tissue; however, the levels of HADH were lower in the latter ($P = 0.015$, $g = 1.40$) (Fig. 5).

Similarly, no significant differences were found in the different WAT markers (AMPK, pATGLSer406, total HSL, pHSLSer565, total HSL/pHSLSer656 ratio or Plin5), with the exception of higher levels of the activated form of AMPK (pAmpkThr172) in McArdle mice ($P = 0.037$, $g = 2.27$) (Fig. 6). This latter finding might suggest a higher activation of lipid metabolism in the WAT of McArdle mice.

### Discussion

The main finding of the present study was that, despite showing a remarkably lower cardiorespiratory fitness ($\dot{V}_{O_2peak}$), patients with McArdle disease exhibited a considerably greater capacity for fat oxidation during endurance exercise [62% or 73% higher MFO rate expressed

in g min$^{-1}$ or relative to fat-free mass (mg kg$^{-1}$ min$^{-1}$), respectively] than their sex- and age-matched healthy referents. Moreover, patients attained FATmax values at near-maximal intensities (94.4 ± 7.2% of $\dot{V}_{O_2peak}$) instead of doing so at moderate intensities, as would be expected for any individual with preserved muscle glycogen availability (i.e. 41.3 ± 9.1% of $\dot{V}_{O_2peak}$ in the

**Table 3. Age, body composition and cardiorespiratory fitness mean (SD) values by group**

| | Patients (n = 9) | Controls (n = 12) | Between-group P value | Between-group Hedge's g |
|---|---|---|---|---|
| Age (years) | 36 (12) | 40 (13) | 0.463 | 0.33 |
| Body mass (kg) | 68.2 (6.5) | 74.2 (9.9) | 0.136 | 0.72 |
| Fat-free mass (kg) | 48.1 (6.0) | 52.8 (8.6) | 0.173 | 0.64 |
| Fat (%) | 25.4 (7.2) | 25.2 (9.0) | 0.948 | 0.03 |
| Peak work capacity (W kg$^{-1}$) | 1.28 (0.36) | 3.23 (0.97) | <0.0001 | 2.66 |
| $\dot{V}_{O_2peak}$ (mL kg$^{-1}$ min$^{-1}$) | 24.7 (4) | 42.5 (11.4) | <0.0001 | 2.07 |

Abbreviations: $\dot{V}_{O_2peak}$, peak oxygen uptake.

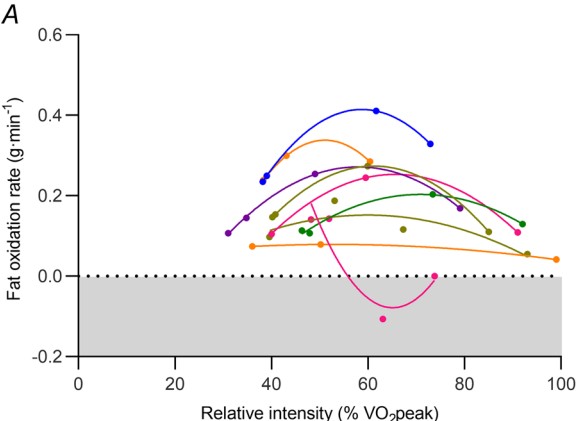

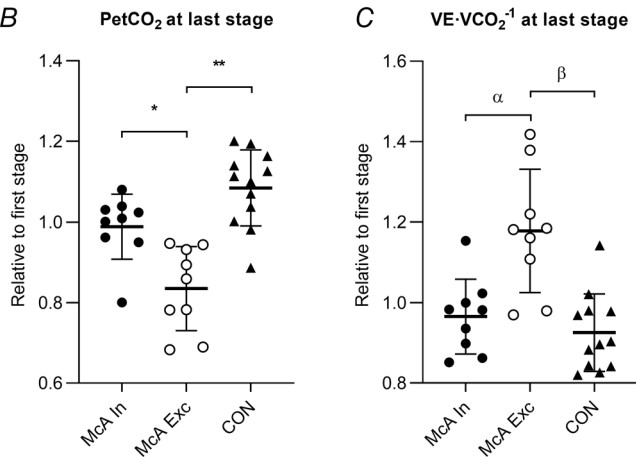

**Figure 2. Data for the patients with McArdle disease excluded from the final analysis of fat oxidation rate as a result of excess hyperpnoea**

These patients (individual values displayed in *A*) showed a decay in fat oxidation rate in the more intense stages of the incremental exercise tests, resulting in a quadratic relationship (instead of the quasi-linear relationship found in the rest of patients, as shown in Fig. 3) between fat oxidation rate and relative exercise intensity [% of peak oxygen uptake ($\dot{V}_{O_2peak}$)].*B* and *C*, comparisons of end-tidal partial pressure of carbon dioxide ($P_{ETCO_2}$) and ventilatory equivalent of carbon dioxide (VE·$\dot{V}_{CO_2}$$^{-1}$) values, respectively, at the last stage of the incremental step test (relative to the first stage) in patients included in the analyses ('McA In') or excluded ('McA Exc'), and in controls ('CON'). Circles and triangles, horizontal bars and error bars show individual data, mean and SD, respectively. *$P$ = 0.005 (Hedge's $g$ = 1.65) for excluded *vs.* included patients; **$P$ < 0.001 ($g$ = 2.44) for excluded patients *vs.* controls; $^{\alpha}P$ = 0.002 ($g$ = 1.68) for excluded *vs.* included patients; $^{\beta}P$ = 0.001 ($g$ = 1.98) for excluded patients *vs.* controls. [Colour figure can be viewed at wileyonlinelibrary.com]

study controls). Analysis of McArdle mice, a faithful model of disease manifestations, revealed no strong evidence for higher levels of molecular markers of fat mobilization/oxidation, especially at the skeletal muscle level.

Previous evidence supports the notion that nutritional status and, particularly, carbohydrate availability, affects fat oxidation rates during exercise, and restricting carbohydrate availability during exercise (e.g. through glycogen depletion induced by prior exercise bouts) has been considered an optimal strategy for maximizing the MFO rate (Achten & Jeukendrup, 2003; Mata et al., 2019; Volek et al., 2016). However, little is known about how glycogen availability affects MFO rate. The present study is the first to our knowledge to evaluate the rate of fat oxidation at different exercise intensities in patients with McArdle disease and to determine the MFO rate using a practical and widely used approach (indirect calorimetry), with this disease representing a unique model of total muscle glycogen unavailability (Santalla et al., 2014). Ørngreen et al. (2009) assessed fat and carbohydrate oxidation in patients with McArdle disease (who had previously passed, as in the present study, the second wind point) during a 40 min exercise bout at a constant submaximal workload (50–60% of $\dot{V}_{O_2peak}$) using a stable isotope technique and indirect calorimetry. In line with our findings, a higher rate (40–70%) of fat oxidation was found for the same workload in patients compared to in healthy subjects. In this regard, a novel finding of the present study is that patients with McArdle disease have a higher capacity for fat oxidation also in absolute terms (i.e. MFO rate) than peers without this condition. Moreover, the workload eliciting the MFO rate was higher in patients compared to in healthy controls, suggesting that the former are better adapted to use fat as fuel to sustain

exercise. The increased mobilization of fatty acids during exercise reported by Ørngreen et al. (2009) together with the larger fat oxidation rates observed through indirect calorimetry in the present study suggest that patients with McArdle disease reach higher MFO rate values than their peers with preserved capacity for muscle glycogen utilization. This phenomenon might be associated with an 'excess' sympathoadrenal response at submaximal intensities aimed at increasing blood flow (and thus availability of blood substrates) to working muscles (Ørngreen et al., 2009; Vissing et al., 1992). Nevertheless, it must be noted that the 'unique' metabolic adaptations in terms of fat oxidation capacity exhibited by these patients are not sufficient to offset the block in glycogenolysis, as reflected by their lower $\dot{V}_{O_2peak}$ and peak work capacity levels compared to normative values, a finding that has been replicated in several studies (Munguía-Izquierdo

et al., 2015; Rae et al., 2010; Salazar-Martínez et al., 2021). Indeed, because of the inherited deficiency of myophosphorylase activity, all patients (irrespective of their activity or fitness levels) have to rely, inevitably, on fat oxidation as a main metabolic pathway. Carbohydrates are much more efficient fuels than fat in terms of ensuring a quick rate of ATP generation per unit of oxygen consumed because they are able to produce a greater ATP yield per unit of oxygen consumption than fat despite the greater ATP production per unit of substrate from the latter (Leverve et al., 2007).

Among the factors that appear to be involved in human fat oxidation capacity, it has been hypothesized that a reduced cycle rate of the tricarboxylic acid (TCA) cycle as a result of the low availability of TCA intermediates (TCAI) may constrain fat oxidation capacity in patients with McArdle disease (Ørngreen et al., 2009;

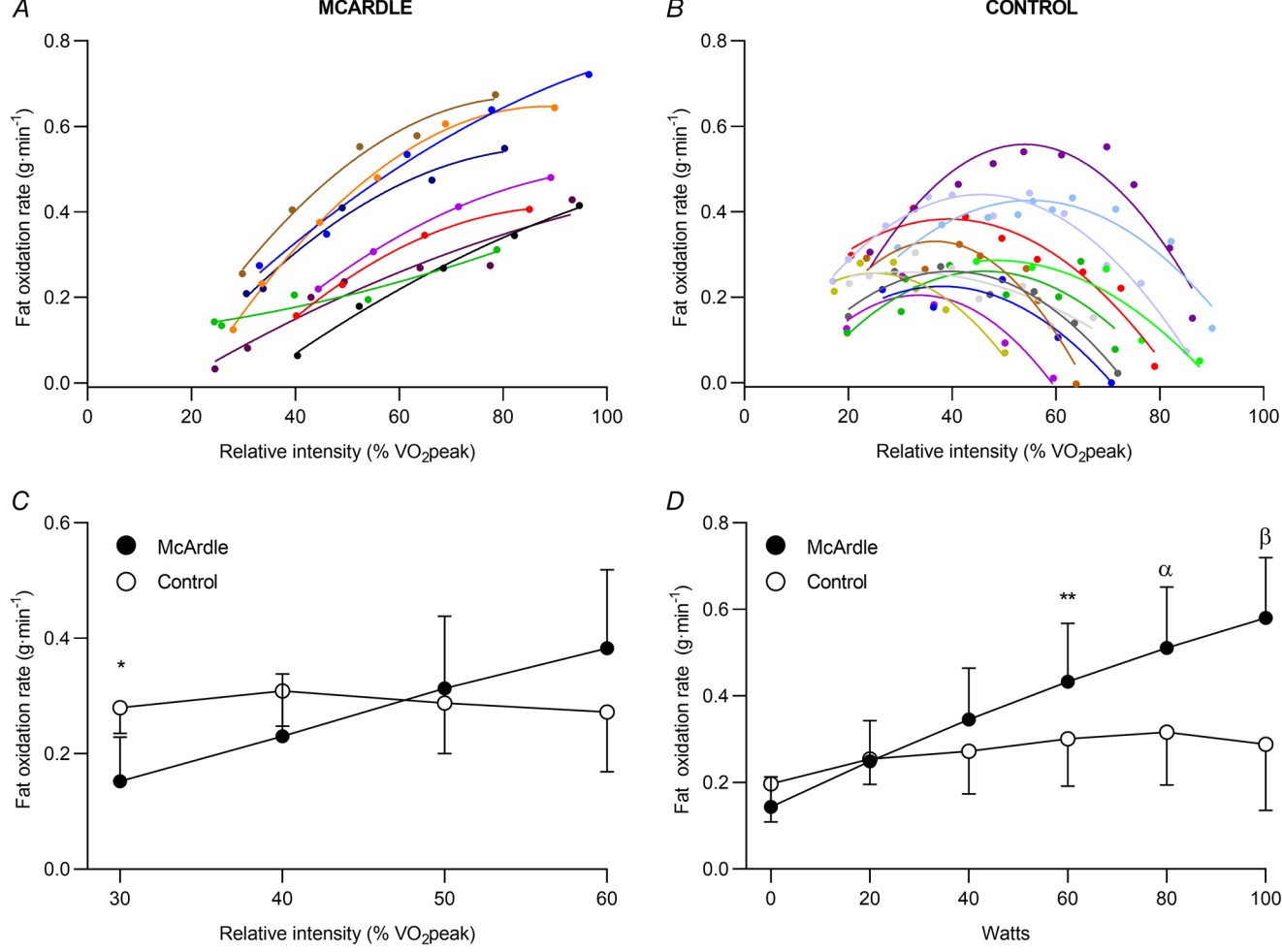

**Figure 3. Individual determinations of the relationship between fat oxidation rate and relative exercise intensity**
A, patients ($n = 9$). B, controls ($n = 12$), as well as mean (SD) values of fat oxidation rate at exercise relative intensities (C) and absolute workload (D), respectively, by group. Circles, horizontal bars and error bars show individual data, mean and SD, respectively Abbreviations: $\dot{V}_{O_2peak}$, peak oxygen uptake. *$P = 0.010$, **$P = 0.026$, $^{\alpha}P = 0.005$ and $^{\beta}P < 0.0001$, respectively, for McArdle vs. controls. [Colour figure can be viewed at wileyonlinelibrary.com]

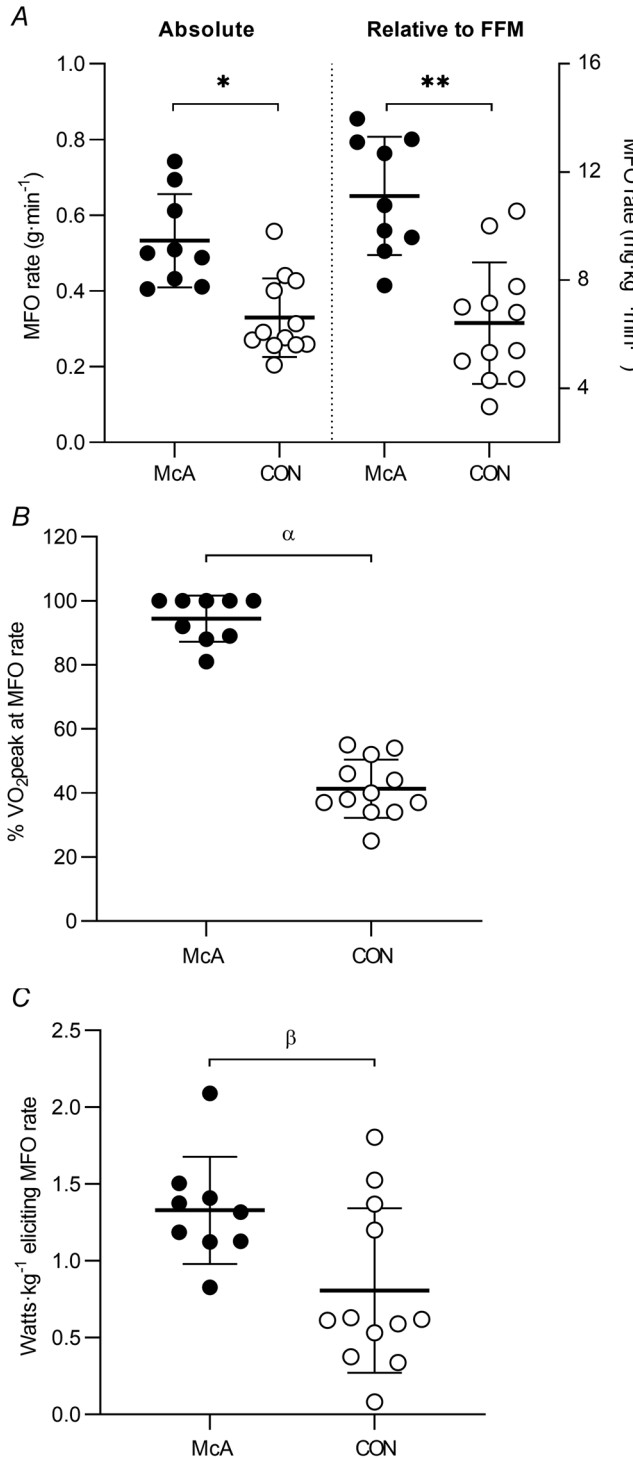

**Figure 4. Fat oxidation in McArdle patients and controls**
Maximal fat oxidation (MFO) rate (*A*), relative exercise intensity (FATmax) (*B*) and workload eliciting MFO (*C*) in McArdle patients (McA, *n* = 9) and controls (CON, *n* = 12). Circles, horizontal bars and error bars show individual data, mean and SD, respectively. Abbreviations: FFM, fat-free mass; $\dot{V}_{O_2peak}$, peak oxygen uptake. *$P$ = 0.001, *$P$ < 0.0001, $^{\alpha}P$ < 0.0001 and $^{\beta}P$ = 0.020, respectively, for McArdle *vs.* controls.

Ørngreen et al., 2015; Sahlin et al., 1995). Low levels of TCAI during exercise have indeed been reported in the skeletal muscle (Sahlin et al., 1995) and plasma (Delaney et al., 2017) of these patients. Delaney et al. (2017) elegantly established a direct relationship between glycogenolysis and TCAI expansion during exercise, supporting the notion that impaired oxidative capacity in McArdle disease is related to a reduced TCAI availability, thereby inducing a metabolic bottleneck for beta-oxidation-derived acetyl-CoA oxidation. In this regard, it is well known that oral sucrose intake (Andersen et al., 2008; Vissing & Haller, 2003b) or ɪ.v. glucose infusion (Haller & Lewis, 1991; Haller & Vissing, 2002) in patients with this condition increases the availability of blood glucose for working fibres, which is subsequently metabolized by skeletal muscle (i.e. bypassing the metabolic block in glycogenolysis), thereby attenuating exercise intolerance at submaximal workloads and partly improving peak work capacity. In any case, carbohydrate ingestion/infusion before exercise does not fully compensate for the inherited blockade in glycogenolysis upstream the uptake of blood glucose by muscle fibres and thus for the impairing effects of this block on the TCA rate (and therefore on oxidative metabolism): indeed, even after ingesting very high amounts of carbohydrates (75 g of fructose) 30 min before exercise, the $\dot{V}_{O_2peak}$ of virtually all patients with McArdle disease (including active physically individuals) are still well below their age-/sex-matched normality value (57% lower on average) (Munguía-Izquierdo et al., 2015). Another factor that has been previously reported to affect MFO rate in humans is training status, with trained/active individuals showing higher MFO rate values than their untrained/inactive referents (Maunder et al., 2018). In this context, an interesting finding of the present study is that, despite the much lower cardio-respiratory fitness of patients with McArdle disease, their MFO rate values were significantly higher than those of healthy controls and, in fact, were comparable to values previously reported in a large cohort (*n* = 1121) of athletes (men and women) of different ages and competitive disciplines, regardless of whether this variable was expressed in 'absolute' units (i.e. 0.53 ± 0.12 g min$^{-1}$ in our patients *vs.* 0.59 ± 0.18 g min$^{-1}$ in athletes) or relative to fat-free mass (11.1 ± 2.19 mg kg$^{-1}$ min$^{-1}$ *vs.* 10.2 ± 2.6 mg kg$^{-1}$ min$^{-1}$) (Randell et al., 2017). Despite some differences in the protocol, age and sex distribution of the sample, this comparison suggests that the MFO rate capacity of McArdle patients can be nearly as high as that of well-trained healthy individuals, which in turn would be able to compensate, at least to some extent, for the inherited blockade in muscle glycogenolysis.

On the other hand, our analysis of a mouse model of the disease revealed no major differences in molecular markers of lipid metabolism/transport that

would theoretically reflect an adaptation favouring fat metabolism in the muscle/WAT. Indeed, the muscle levels of HADH, a mitochondrial fatty acid $\beta$-oxidation enzyme, were lower in McArdle mice compared to in WT mice. Analysis of WAT, however, revealed significantly higher levels of activated AMPK in McArdle mice (with a large effect size), which is a key regulator of fat metabolism, by regulating fatty acid transport into the mitochondria or by regulating PGC-1$\alpha$ signalling (Thomson & Winder, 2009).

We also found that MFO rate occurred at near maximal exercise capacity ($\sim$ 40% *vs.* 95% of $\dot{V}_{O_2peak}$, respectively) in patients with McArdle disease compared to healthy controls, in whom the MFO rate occurred at moderate-intensity exercise. This finding suggests that MFO rate could be a main limiting factor of endurance exercise performance in patients with McArdle disease. It would be interesting to explore the effect of different training or nutritional interventions on MFO rate and exercise tolerance in these patients in a future analysis. In this regard, there is evidence that regular aerobic exercise can improve the $\dot{V}_{O_2peak}$ of patients with McArdle disease (Haller et al., 2006; Maté-Muñoz et al., 2007; Porcelli et al., 2016; Santalla et al., 2022) and previous research

has assessed the effects of acute (pre-exercise) oral administration of branched-chain amino acids (MacLean et al., 1998) or sucrose (Andersen et al., 2008; Vissing & Haller, 2003b) or of i.v. infusion of glucose (Haller & Lewis, 1991; Haller & Vissing, 2002), triglycerides (Haller & Lewis, 1991), nicotinic acid (a blocker of lipolysis) and a lipid emulsion (soybean oil) (Andersen et al., 2009), as well as fasting during the previous 38 h (Carroll et al., 1979), on the exercise tolerance of these patients. However, to the best of our knowledge, no previous study has assessed the effect of these interventions on MFO rate, which might be addressed in future research.

Some methodological limitations of the present study should be acknowledged. Although we can confidently assert that patients with McArdle disease serve as a model of total glycogen unavailability, we did not assess pre-exercise muscle glycogen levels in healthy controls, which can affect fat oxidation rates. Similarly, muscle glycogen levels were not determined in WT and McArdle mice. Concerning the later, however, previous research by our group has shown that glycogen levels are 27 higher on average in the gastrocnemius muscle of McArdle mice (48–58 mg g$^{-1}$ of tissue) compared to in WT controls (1.6–2.6 mg g$^{-1}$) (Nogales-Gadea et al., 2012).

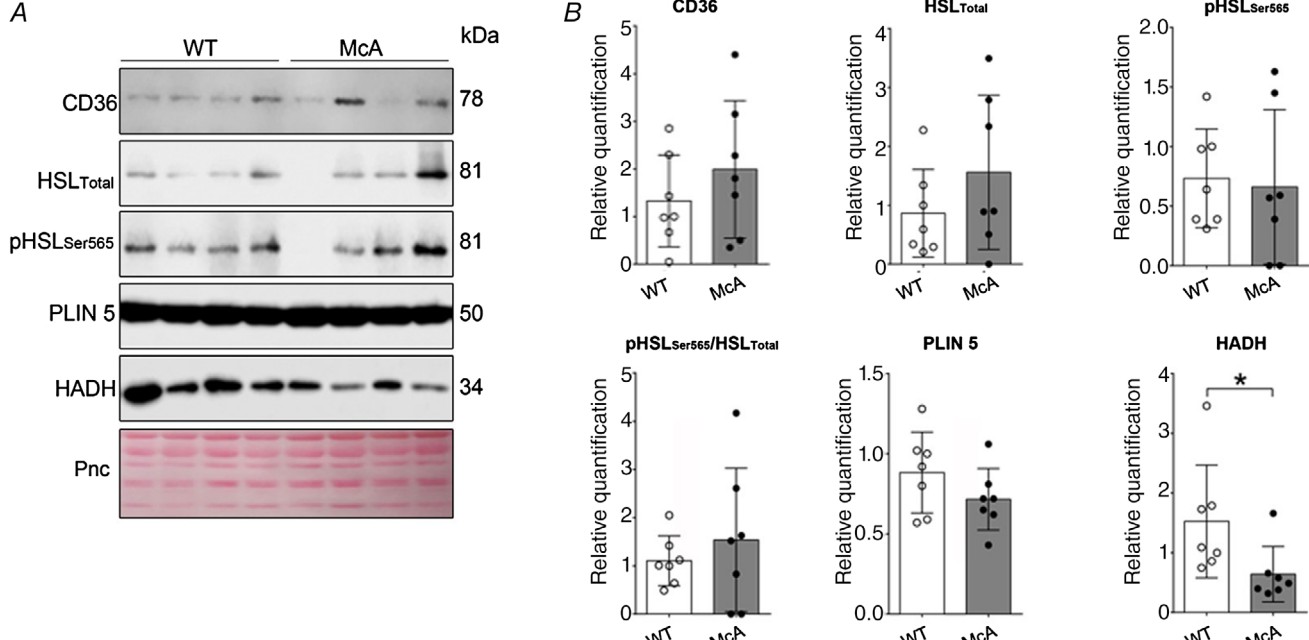

**Figure 5. Protein levels indicative of lipid mobilization and catabolism in skeletal muscle tissue in McArdle (p.R50*/p.R50*. McA) and wild-type (WT) male mice**
Western blot images shown in (*A*) and results expressed as mean (SD), as well as individual values (*n* = 6 or 7 per group) shown in (*B*). Abbreviations: CD36, transmembrane glycoprotein cluster of differentiation 36; HADH, 3-hydroxyacyl-CoA dehydrogenase; HSL, hormone-sensitive lipase; pHSLSer565, hormone-sensitive lipase phosphorylated at serine 656; Plin5, perilipin 5; Pnc, Ponceau S; *P = 0.015 for the comparison of McArdle *vs.* WT mice. The remaining *P* values for the comparison of McArdle *vs.* WT mice were: CD36, *P* = 0.441; total HSL, *P* = 0.158; pHSLSer565, *P* = 0.795; ratio of pHSLSer565/total HSL, *P* = 0.704; and Plin5, *P* = 0.373. All the data correspond to the gastrocnemius muscles. [Colour figure can be viewed at wileyonlinelibrary.com]

An additional limitation of the mouse experiments is the small sample size and variability of the results, which precludes drawing solid conclusions regarding the molecular markers we measured. On the other hand, although all participants were tested after an overnight fast, dietary intake was not controlled for in the day(s) prior to the testing, which could be a confounding factor for substrate oxidation. It is also worth noting that the study participants were assessed only once; in this effect, determination of substrate oxidation rates by gas exchange measurements as we did here shows large day-by-day variability (Chrzanowski-Smith et al., 2020), which might have confounded, at least partly, our results. Moreover, this technique relies on the assumption that expired gas exchange data correspond to gas exchange at the tissue level (Frayn, 1983; Jeukendrup & Wallis, 2005). In healthy individuals, the increased glycolytic flux during high-intensity exercise raises [H$^+$], which is buffered by HCO$_3^-$, thereby resulting in an excess of non-oxidative CO$_2$ that will be removed throughout hyperpnoea. Accordingly, it has been suggested that calculations of fat oxidation rates using indirect calorimetry will be flawed at high exercise intensities (i.e. ≥75% of $\dot{V}_{O_2peak}$) (Jeukendrup & Wallis, 2005). In this regard, although the production of non-oxidative CO$_2$ in patients with McArdle disease could be considered negligible as a result of the complete block of muscle glycogen breakdown (Ørngreen et al., 2015), we registered an excess $\dot{V}_{CO_2}$ combined with hyperpnoea during the last stage of the steady-state incremental protocol in almost half of the patients of our initial sample. We therefore hypothesized that, because the contribution of [H$^+$] buffering must be obviously discarded, the excess $\dot{V}_{CO_2}$ was the result of an increase in alveolar respiration induced by hyperpnoea. In this line, Hagberg et al. (1982) reported the occurrence of hyperpnoea during intense exercise in patients with McArdle disease without a concomitant increase in plasma [H$^+$], but with a marked decline in $P_{ETCO_2}$ (which is a proxy for blood partial pressure of CO$_2$). The excessive clearance of circulating CO$_2$ through breathing might have thus increased RER artificially in some patients and influenced the calculations of fat oxidation rate because the assumption of equivalence between breathing and tissue gas exchange of indirect calorimetry might have been breached. On the other hand, in light of the lower $\dot{V}_{O_2peak}$ values of the patients excluded from final analyses of fat oxidation rate compared to those finally included (18.1 ± 6.4 *vs.* 24.8 ± 4 mL min$^{-1}$ kg$^{-1}$, *g* = 1.26, *P* = 0.017), we can hypothesize that aerobic fitness is linked to the magnitude of the hyperpneic response to exercise in McArdle disease, such that the fittest individuals may experience milder hyperpnoea.

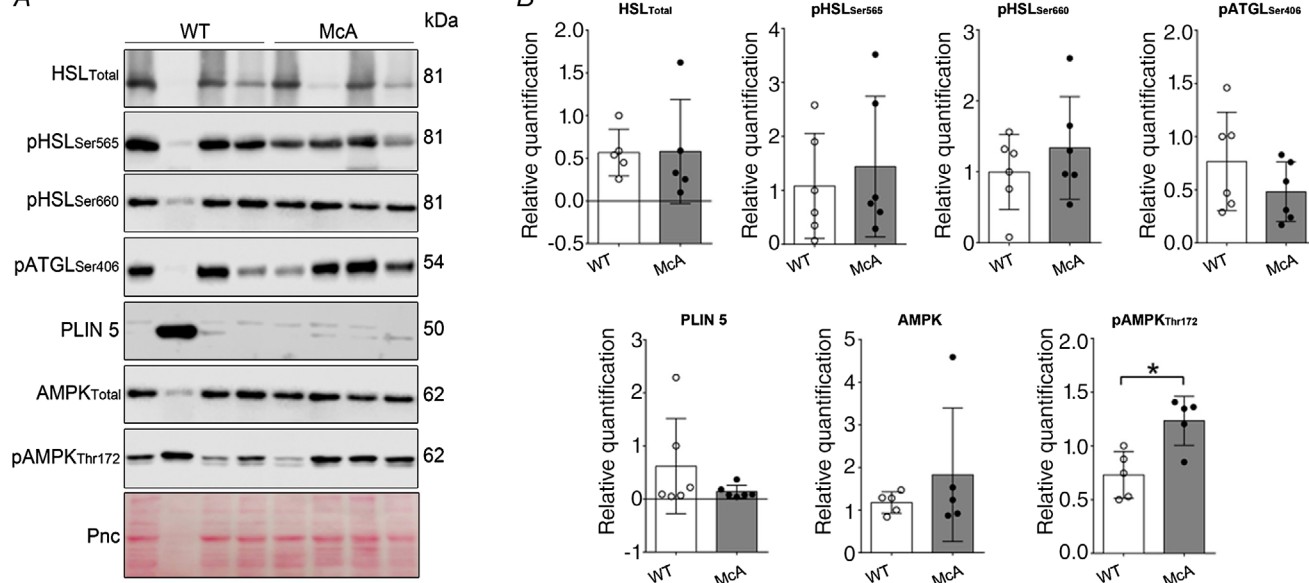

**Figure 6. Protein levels indicative of lipid mobilization and catabolism in the white adipose tissue of McArdle (McA, p.R50*/p.R50*) or wild-type (WT) male mice**
Western blot images shown in (*A*) and results expressed as mean (SD), as well as individual values (*n* = 5 or 6 per group) shown in (*B*). Abbreviations: AMPK, AMP-activated protein kinase; HSL$_{Total}$, hormone-sensitive lipase (total levels); pATGL$_{Ser406}$, adipose triglyceride lipase (ATGL) phosphorylated at serine 406; pAMPK$_{Thr172}$, AMPK phosphorylated at threonine 172; pHSL$_{Ser565}$, HSL phosphorylated at serine 565; pHSL$_{Ser660}$, HSL phosphorylated at serine 660; Plin5, perilipin 5; Pnc, Ponceau S. *$P$ = 0.037 for the comparison of McArdle *vs.* WT mice. The remaining $P$ values for the comparison of McArdle *vs.* WT mice were: AMPK, $P$ = 0.471; pATGLSer406, $P$ = 0.818; pHSLSer565, $P$ = 0.562; pHSLSer660, $P$ = 0.689; Plin 5, $P$ = 0.522; and total HSL, $P$ = 0.928. [Colour figure can be viewed at wileyonlinelibrary.com]

This being said, considering the large number of patients presenting with marked hyperpnoea (i.e. 50% of the initial sample), our data must be interpreted with caution and future studies exploring substrate oxidation using stable isotopes at different exercise intensities in patients with McArdle disease are warranted. This technique would allow us to discern whether the high fat oxidation capacity of these patients is also extensible to those with exercise-induced hyperpnoea or whether the maximal exercise capacity of this patient population is also sustained by oxidation of other substrates (e.g. branched chain amino acids, or glucose mobilized from the liver or from glycophagy pathways). Finally, we could not perform molecular determinations at the muscle or WAT level in patients because of ethical constraints, although we consider that this was compensated by the measurements in the disease model of this disease, which closely mimics the main features of patients (Nogales-Gadea et al., 2012).

In conclusion, physically active patients with McArdle disease and non-severe clinical affectation exhibit an exceptional MFO rate capacity, superior to that of their healthy counterparts. Moreover, as opposed to healthy controls in which MFO rate occurred at moderate-intensity workloads, the MFO rate was attained by near-maximal workloads in these patients. These findings support the notion that tolerance to intense exercise in patients with McArdle disease primarily relies on MFO rate as a potential adaptation to compensate (at least partly) for the inherited block in glycogenolysis.

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

## Additional information

### Data availability statement

The data that support the findings of this study are available from the corresponding author upon reasonable request.

### Competing interests

The authors declare that they have no competing interests.

### Author contributions

C.R.L., A.S., P.L.V., A.L. and I.A. were responsible for conceptualization and design of the study. C.R.L., A.R., M.V.S., A.S., I.R.G., T.P. were responsible for acquisition of data. C.R.L., A.R., M.V.S., A.S., I.R.G. were responsible for analysis and interpretation of data. C.R.L., A.S. were responsible for writing the original draft. C.R.L., A.S., P.L.V., I.R.G., T.P, A.L. and I.A. were responsible for reviewing and editing. A.L. and I.A. were responsible for supervision. All authors have approved the final version of the manuscript and agree to be accountable for all aspects of the work. All persons designated as authors qualify for authorship, and all those who qualify for authorship are listed.

### Funding

Research by the IA and CR-L group is funded by the Biomedical Research Networking Centre on Frailty and Healthy Aging (CIBERFES, CB16/10/00314 and CB16/10/00477). IR-G is supported by a postdoctoral contract from Universidad de Castilla–La Mancha (2021/5937). PLV is supported by a Sara Borrell contract from Instituto de Salud Carlos III (CD21/00138). Research by AL and TP is funded by the Spanish Ministry of Economy and Competitiveness and Fondos FEDER (PI18/00139 and PI19/01313, respectively).

### Acknowledgements

We thank Dr Kenneth McCreath for his editorial assistance.

### Keywords

anaplerotic, fatty acids, glycogen depletion, glycogen store disease, lactate, muscle fatigue, substrate oxidation, tricarboxylic acid cycle

## Supporting information

Additional supporting information can be found online in the Supporting Information section at the end of the HTML view of the article. Supporting information files available:

**Statistical Summary Document**
**Peer Review History**

