## [Peer Review History · The Journal of Physiology]

Muscle glycogen unavailability and fat oxidation rate during exercise: Insights from McArdle disease

Carlos Rodriguez-Lopez, Alfredo Santalla, Pedro L Valenzuela, Alberto Real-Martinez, Monica Villareal-Salazar, Irene Rodríguez-Gómez, Tomàs Pinós, Alejandro Lucia, and Ignacio Ara

DOI: 10.1113/JP283743

Corresponding author(s): Ignacio Ara (ignacio.ara@uclm.es)

The following individual(s) involved in review of this submission have agreed to reveal their identity: Joachim Nielsen (Referee #2)

Review Timeline:

Submission Date:	17-Aug-2022
Editorial Decision:	27-Sep-2022
Revision Received:	21-Oct-2022
Accepted:	31-Oct-2022

Senior Editor: Michael Hogan

Reviewing Editor: Javier Gonzalez

Transaction Report:

Dear Professor Ara,

Re: JP-RP-2022-283743 "Muscle glycogen unavailability and fat oxidation rate during exercise: Insights from McArdle disease" by Carlos Rodriguez-Lopez, Alfredo Santalla, Pedro L Valenzuela, Alberto Real-Martinez, Monica Villareal-Salazar, Irene Rodriguez-Gomez, Tomàs Pinós, Alejandro Lucia, and Ignacio Ara

Thank you for submitting your manuscript to The Journal of Physiology. It has been assessed by a Reviewing Editor and by 2 expert Referees and I am pleased to tell you that it is considered to be acceptable for publication following satisfactory revision.

The reports are copied at the end of this email. Please address all of the points and incorporate all requested revisions, or explain in your Response to Referees why a change has not been made.

NEW POLICY: In order to improve the transparency of its peer review process The Journal of Physiology publishes online as supporting information the peer review history of all articles accepted for publication. Readers will have access to decision letters, including all Editors' comments and referee reports, for each version of the manuscript and any author responses to peer review comments. Referees can decide whether or not they wish to be named on the peer review history document.

Authors are asked to use The Journal's premium BioRender (<https://biorender.com/>) account to create/redraw their Abstract Figures. Information on how to access The Journal's premium BioRender account is here: <https://physoc.onlinelibrary.wiley.com/journal/14697793/biorender-access> and authors are expected to use this service. This will enable Authors to download high-resolution versions of their figures. The link provided should only be used for the purposes of this submission. Authors will be charged for figures created on this premium BioRender account if they are not related to this manuscript submission.

I hope you will find the comments helpful and have no difficulty returning your revisions within 4 weeks.

Your revised manuscript should be submitted online using the links in Author Tasks Link Not Available.

Any image files uploaded with the previous version are retained on the system. Please ensure you replace or remove all files that have been revised.

REVISION CHECKLIST:

- Article file, including any tables and figure legends, must be in an editable format (eg Word)
- Abstract figure file (see above)
- Statistical Summary Document
- Upload each figure as a separate high quality file
- Upload a full Response to Referees, including a response to any Senior and Reviewing Editor Comments;
- Upload a copy of the manuscript with the changes highlighted.

- A potential 'Cover Art' file for consideration as the Issue's cover image;
- Appropriate Supporting Information (Video, audio or data set https://jp.msubmit.net/cgi-bin/main.plex?form_type=display_requirements#supp).

To create your 'Response to Referees' copy all the reports, including any comments from the Senior and Reviewing Editors, into a Word, or similar, file and respond to each point in colour or CAPITALS and upload this when you submit your revision.

I look forward to receiving your revised submission.

If you have any queries please reply to this email and staff will be happy to assist.

Yours sincerely,

Michael C. Hogan
Senior Editor
The Journal of Physiology
<https://jp.msubmit.net>
<http://jp.physoc.org>
The Physiological Society
Hodgkin Huxley House
30 Farringdon Lane
London, EC1R 3AW
UK
<http://www.physoc.org>
<http://journals.physoc.org>

REQUIRED ITEMS:

-Author photo and profile. First (or joint first) authors are asked to provide a short biography (no more than 100 words for one author or 150 words in total for joint first authors) and a portrait photograph. These should be uploaded and clearly labelled with the revised version of the manuscript. See Information for Authors for further details.

-You must start the Methods section with a paragraph headed Ethical Approval. If experiments were conducted on humans confirmation that informed consent was obtained, preferably in writing, that the studies conformed to the standards set by the latest revision of the Declaration of Helsinki, and that the procedures were approved by a properly constituted ethics committee, which should be named, must be included in the article file. If the research study was registered (clause 35 of the Declaration of Helsinki) the registration database should be indicated, otherwise the lack of registration should be noted as an exception (e.g. The study conformed to the standards set by the Declaration of Helsinki, except for registration in a database.). For further information see: <https://physoc.onlinelibrary.wiley.com/hub/human-experiments>

-The Journal of Physiology funds authors of provisionally accepted papers to use the premium BioRender site to create high resolution schematic figures. Follow this link and enter your details and the manuscript number to create and download figures. Upload these as the figure files for your revised submission. If you choose not to take up this offer we require figures to be of similar quality and resolution. If you are opting out of this service to authors, state this in the Comments section on the Detailed Information page of the submission form. The link provided should only be used for the purposes of this submission. Authors will be charged for figures created on this premium BioRender account if they are not related to this manuscript submission.

-Please upload separate high-quality figure files via the submission form.

-You must upload original, uncropped western blot/gel images (including controls) if they are not included in the manuscript. This is to confirm that no inappropriate, unethical or misleading image manipulation has occurred <https://physoc.onlinelibrary.wiley.com/hub/journal-policies#imagmanip> These should be uploaded as 'Supporting information for review process only'. Please label/highlight the original gels so that we can clearly see which sections/lanes have been used in the manuscript figures.

-Your paper contains Supporting Information of a type that we no longer publish. Any information essential to an understanding of the paper must be included as part of the main manuscript and figures. The only Supporting Information that we publish are video and audio, 3D structures, program codes and large data files. Your revised paper will be returned to you if it does not adhere to our Supporting Information Guidelines

-A Statistical Summary Document, summarising the statistics presented in the manuscript, is required upon revision. It must be on the Journal's template, which can be downloaded from the link in the Statistical Summary Document section here: https://jp.msubmit.net/cgi-bin/main.plex?form_type=display_requirements#statistics

-Papers must comply with the Statistics Policy https://jp.msubmit.net/cgi-bin/main.plex?form_type=display_requirements#statistics

In summary:

-If n {less than or equal to} 30, all data points must be plotted in the figure in a way that reveals their range and distribution. A bar graph with data points overlaid, a box and whisker plot or a violin plot (preferably with data points included) are acceptable formats.

-If $n > 30$, then the entire raw dataset must be made available either as supporting information, or hosted on a not-for-profit repository e.g. FigShare, with access details provided in the manuscript.

-' n ' clearly defined (e.g. x cells from y slices in z animals) in the Methods. Authors should be mindful of pseudoreplication.

-All relevant ' n ' values must be clearly stated in the main text, figures and tables, and the Statistical Summary Document (required upon revision)

-The most appropriate summary statistic (e.g. mean or median and standard deviation) must be used. Standard Error of the Mean (SEM) alone is not permitted.

-Exact p values must be stated. Authors must not use 'greater than' or 'less than'. Exact p values must be stated to three significant figures even when 'no statistical significance' is claimed.

-Statistics Summary Document completed appropriately upon revision

-Please include an Abstract Figure. The Abstract Figure is a piece of artwork designed to give readers an immediate understanding of the research and should summarise the main conclusions. If possible, the image should be easily 'readable' from left to right or top to bottom. It should show the physiological relevance of the manuscript so readers can assess the importance and content of its findings. Abstract Figures should not merely recapitulate other figures in the manuscript. Please try to keep the diagram as simple as possible and without superfluous information that may distract from the main conclusion(s). Abstract Figures must be provided by authors no later than the revised manuscript stage and should be uploaded as a separate file during online submission labelled as File Type 'Abstract Figure'. Please ensure that you include the figure legend in the main article file. All Abstract Figures should be created using BioRender. Authors should use The Journal's premium BioRender account to export high-resolution images. Details on how to use and access the premium account are included as part of this email.

EDITOR COMMENTS

Reviewing Editor:

The current manuscript has been reviewed by two experts in the field. Both highlight the potential for this manuscript to enhance knowledge of exercise metabolism. There are however, some important comments that require careful consideration by the authors and appropriate amendments to the manuscript before it can be considered any further.

REFEREE COMMENTS

Referee #1:

Thank you for the opportunity to review this paper on such a fascinating population. The study itself demonstrates that maximal fat oxidation occurs at near maximal exercise capacity in McArdle's patients, and the absolute rates of maximal fat oxidation that they exhibit are similar to well-trained disease-free individuals.

Below are my comments that the authors should consider in a revised manuscript.

Line 4 - remove "no"?

Line 106 - should be higher MFO 'rate'. This needs to be amended throughout.

Line 151-206 - Was there any dietary control in the day(s) prior to the fatmax test? Or even information on habitual diet and whether this was similar between patients and controls.

Line 218 - Why was the gastrocnemius muscle chosen? It would be useful to briefly justify the selection.

Line 299 - Report age in whole numbers in Table 1.

Figure 3 - panel C, the -1 in watts.kg should be superscript.

Line 329 - what do you mean by steady-state? Resting? Can this be clarified please.

Line 335 - How have you determined that there is 'voids full of glycogen granules'? The perfect solution here would be to stain using PAS or if you have access to a glycogen antibody (e.g. <https://pubmed.ncbi.nlm.nih.gov/24204959/>) and provide a representative image. This would add clarity especially for a non-expert, and the images would be very interesting for a reader who has a good understanding of microscopy. It could also be possible to provide a semi-quantitative assessment of glycogen content; this would be useful as to address the limitation of a lack of glycogen quantification.

Line 337 - need to define what panel C in Figure 4 illustrates.

Line 391 - Suggest to re-word this. Only Orngreen et al. (2009) have measured plasma FFA concentrations to show that there was increased mobilization of fatty acids during exercise. This was not measured in the present study.

Line 412-420 - Initially it is stated that the impaired oxidative capacity (and lower work capacity) in patients is related to the compromised rate of the TCA cycle. But then it is explained that carbohydrate feeding/infusion can overcome this intolerance. Presumably then, the TCA cycle isn't compromised per se in patients, since glucose oxidation would require this to be functional in order to improve work capacity. So is the TCA cycle truly compromised in patients or not?

Line 439-440 - Can you expand on this adaptation from a mechanistic perspective? Perhaps there is greater CD36 located at the mitochondria as well as the plasma membrane in patients, in order to facilitate fatty acid entry into the mitochondria (<https://pubmed.ncbi.nlm.nih.gov/16670153/>).

Line 446 - Just clarify in this sentence that you are referring to patients rather than healthy individuals.

Whilst I appreciate the effort the authors have gone to discuss the limitations, I believe there should also be a comment on the use of a single test, in light of data suggesting that there is poor day-to-day reliability in measures of fatmax (e.g. Chrzanowski-Smith et al., 2020, EJAP). How may have this affected the results of the study? Related to this, it would also be pertinent to comment on the lack of dietary control prior to the exercise testing.

Referee #2:

In this study, the authors investigated the effect of exercise intensity on maximal fat oxidation rate in physically active McArdle patients and healthy controls. This was combined with measures of molecular markers of fat metabolism in skeletal muscle and white adipose tissue from McArdle and wild type mice. They find that McArdle patients achieve very high and maximal fat oxidation rates at near maximal exercise intensity, which suggest that in the condition of blocked glycogenolysis the maximal fat oxidation rate can be a limiting factor for exercise tolerance. The study is original and very well conducted and will impact the area of research by linking glycogen availability to very high fat oxidation rates even in non-athletes. Their conclusion is valid. I have two major comments and a few minor comments. The first major comment is about the reason for excluding 9 patients from the analyses. Although the curves of fat oxidation rate versus exercise intensity look different from the included curves and the mean values of PetCO₂ at VO₂ peak and the VE-VCO₂-1 at VO₂ peak are different, the reason for excluding each of the participants is missing. The second major comment is about data handling and interpretation of the findings from the samples collected from wild type and McArdle mice.

Major comments:

1. The manuscript can be improved by including more details on the reason for the exclusion of data from some McArdle patients (line 198 and 288). The authors should also consider moving the figures presented in the supplemental file to the main article text and present much more detail on how the potentially biased values were identified and if the PetCO₂ at VO₂ peak or the VE-VCO₂-1 at VO₂ peak could be used as a criterion for exclusion. In supplemental figure 3B and 3C there seems to be an overlap of the SD values between groups, which could indicate that some of the excluded McArdle patients had values close to the values of included patients and the controls. It would therefore be clearer if the individual values (PetCO₂ at VO₂ peak and the VE-VCO₂-1 at VO₂ peak) are shown in figure S3B and S3C with specific comments on why some patients (if any) with normal PetCO₂ at VO₂ peak and VE-VCO₂-1 at VO₂ peak values are excluded. These details are important for the understanding of why some data are excluded and the potential confounding role of hyperpnea.

2. The collection of muscle samples from wild-type and McArdle mice is an important addition to the manuscript. However, given the relative low sample size and the high variability in data between muscles the findings remain rather inconclusive. Therefore, the manuscript could be improved by a presentation and discussion of data, which include information about how certain the findings are. This could be confidence intervals for the differences between groups and a discussion on whether the findings are physiological relevant and/or if the results are too uncertain to make any clear conclusion. The inclusion of immunofluorescence analyses of CD36 (Fig 4C) seems very preliminary (only qualitative observations are provided) and should therefore either be removed from the manuscript or the images should be analyzed quantitatively.

Minor comments:

Line 40: The wording of key point 5 seems wrong. Should it read "An animal model revealed no..."?

Line 58: It should be clearly stated that these data refer to data obtained from mice: "No between-group differences were found in molecular markers".

Line 152: If possible, more detailed information on the overnight fast should be included. Within the current statement participants could be fasting for very different durations depending on the specific time of day for the testing and their last meal the evening before. Some may have been fasting for 12-14 hours while others for only 7-8 hours. Since fasting increases fat oxidation (PMID: 9931180) any difference in the fasting duration could affect the results. Therefore, the manuscript could be improved by more specific information on the fasting duration and/or discuss how this can have affected

the results.

Line 181: Correct wording "and/or"?

Fig 4C is not mentioned in the legend.

END OF COMMENTS

Confidential Review

17-Aug-2022

Dear Professor Ara,

Re: JP-RP-2022-283743 "Muscle glycogen unavailability and fat oxidation rate during exercise: Insights from McArdle disease" by Carlos Rodriguez-Lopez, Alfredo Santalla, Pedro L Valenzuela, Alberto Real-Martinez, Monica Villareal-Salazar, Irene Rodriguez-Gomez, Tomàs Pinós, Alejandro Lucia, and Ignacio Ara

Thank you for submitting your manuscript to The Journal of Physiology. It has been assessed by a Reviewing Editor and by 2 expert Referees and I am pleased to tell you that it is considered to be acceptable for publication following satisfactory revision.

The reports are copied at the end of this email. Please address all of the points and incorporate all requested revisions or explain in your Response to Referees why a change has not been made.

NEW POLICY: In order to improve the transparency of its peer review process The Journal of Physiology publishes online as supporting information the peer review history of all articles accepted for publication. Readers will have access to decision letters, including all Editors' comments and referee reports, for each version of the manuscript and any author responses to peer review comments. Referees can decide whether or not they wish to be named on the peer review history document.

Authors are asked to use The Journal's premium BioRender (<https://biorender.com/>) account to create/redraw their Abstract Figures. Information on how to access The Journal's premium BioRender account is here: <https://physoc.onlinelibrary.wiley.com/journal/14697793/biorender-access> and authors are expected to use this service. This will enable Authors to download high-resolution versions of their figures. The link provided should only be used for the purposes of this submission. Authors will be charged for figures created on this premium BioRender account if they are not related to this manuscript submission.

I hope you will find the comments helpful and have no difficulty returning your revisions within 4 weeks.

Your revised manuscript should be submitted online using the links in Author Tasks <https://jp.msubmit.net/cgi-bin/main.plex?el=A1JS4FNK7A2bIF3F5A9ftd3UvTHgsl3UcMevO1Z0UCpgZ>.

Any image files uploaded with the previous version are retained on the system. Please ensure you replace or remove all files that have been revised.

REVISION CHECKLIST:

- Article file, including any tables and figure legends, must be in an editable format (eg Word)
- Abstract figure file (see above)
- Statistical Summary Document

- Upload each figure as a separate high quality file
- Upload a full Response to Referees, including a response to any Senior and Reviewing Editor Comments;
- Upload a copy of the manuscript with the changes highlighted.

- A potential 'Cover Art' file for consideration as the Issue's cover image;
- Appropriate Supporting Information (Video, audio or data set https://jp.msubmit.net/cgi-bin/main.plex?form_type=display_requirements#supp).

To create your 'Response to Referees' copy all the reports, including any comments from the Senior and Reviewing Editors, into a Word, or similar, file and respond to each point in colour or CAPITALS and upload this when you submit your revision.

I look forward to receiving your revised submission.

If you have any queries, please reply to this email and staff will be happy to assist.

Yours sincerely,

Michael C. Hogan
Senior Editor
The Journal of Physiology
<https://jp.msubmit.net>
<http://jp.physoc.org>
The Physiological Society
Hodgkin Huxley House
30 Farringdon Lane
London, EC1R 3AW
UK
<http://www.physoc.org>
<http://journals.physoc.org>

REQUIRED ITEMS:

-Author photo and profile. First (or joint first) authors are asked to provide a short biography (no more than 100 words for one author or 150 words in total for joint first authors) and a portrait photograph. These should be uploaded and clearly labelled with the revised version of the manuscript. See Information for Authors for further details.

Done.

-You must start the Methods section with a paragraph headed Ethical Approval. If experiments were conducted on humans' confirmation that informed consent was obtained, preferably in writing, that the studies conformed to the standards set by the latest revision of the

Declaration of Helsinki, and that the procedures were approved by a properly constituted ethics committee, which should be named, must be included in the article file. If the research study was registered (clause 35 of the Declaration of Helsinki) the registration database should be indicated, otherwise the lack of registration should be noted as an exception (e.g. The study conformed to the standards set by the Declaration of Helsinki, except for registration in a database.). For further information see: <https://physoc.onlinelibrary.wiley.com/hub/human-experiments>

Done. The first paragraph of the Methods section has been modified accordingly.

The Journal of Physiology funds authors of provisionally accepted papers to use the premium BioRender site to create high resolution schematic figures. Follow this link and enter your details and the manuscript number to create and download figures. Upload these as the figure files for your revised submission. If you choose not to take up this offer we require figures to be of similar quality and resolution. If you are opting out of this service to authors, state this in the Comments section on the Detailed Information page of the submission form. The link provided should only be used for the purposes of this submission. Authors will be charged for figures created on this premium BioRender account if they are not related to this manuscript submission.

Done.

-Please upload separate high-quality figure files via the submission form.

Done.

-You must upload original, uncropped western blot/gel images (including controls) if they are not included in the manuscript. This is to confirm that no inappropriate, unethical or misleading image manipulation has occurred
<https://physoc.onlinelibrary.wiley.com/hub/journal-policies#imagmanip> These should be uploaded as 'Supporting information for review process only'. Please label/highlight the original gels so that we can clearly see which sections/lanes have been used in the manuscript figures.

Done.

-Your paper contains Supporting Information of a type that we no longer publish. Any information essential to an understanding of the paper must be included as part of the main manuscript and figures. The only Supporting Information that we publish are video and audio, 3D structures, program codes and large data files. Your revised paper will be returned to you if it does not adhere to our Supporting Information Guidelines

Done. Supporting Information of the previous version has now been moved to the main manuscript. As such, former Supplementary files 1, 2 and 3 now appear as Table 1, Table 2 and Figure 2, respectively.

-A Statistical Summary Document, summarising the statistics presented in the manuscript, is required upon revision. It must be on the Journal's template, which can be downloaded from the link in the Statistical Summary Document section here: https://jp.msubmit.net/cgi-bin/main.plex?form_type=display_requirements#statistics

Done.

-Papers must comply with the Statistics Policy https://jp.msubmit.net/cgi-bin/main.plex?form_type=display_requirements#statistics

Journal's Statistics Policy has been fulfilled.

In summary:

-If n {less than or equal to} 30, all data points must be plotted in the figure in a way that reveals their range and distribution. A bar graph with data points overlaid, a box and whisker plot or a violin plot (preferably with data points included) are acceptable formats.

-If $n > 30$, then the entire raw dataset must be made available either as supporting information, or hosted on a not-for-profit repository e.g. FigShare, with access details provided in the manuscript.

-' n ' clearly defined (e.g. x cells from y slices in z animals) in the Methods. Authors should be mindful of pseudoreplication.

-All relevant ' n ' values must be clearly stated in the main text, figures and tables, and the Statistical Summary Document (required upon revision)

-The most appropriate summary statistic (e.g. mean or median and standard deviation) must be used. Standard Error of the Mean (SEM) alone is not permitted.

-Exact p values must be stated. Authors must not use 'greater than' or 'less than'. Exact p values must be stated to three significant figures even when 'no statistical significance' is claimed.

-Statistics Summary Document completed appropriately upon revision

All statistical data are now presented as per journal guidelines.

-Please include an Abstract Figure. The Abstract Figure is a piece of artwork designed to give readers an immediate understanding of the research and should summarise the main conclusions. If possible, the image should be easily 'readable' from left to right or top to bottom. It should show the physiological relevance of the manuscript so readers can assess the importance and content of its findings. Abstract Figures should not merely recapitulate other figures in the manuscript. Please try to keep the diagram as simple as possible and without superfluous information that may distract from the main conclusion(s). Abstract Figures must be provided by authors no later than the revised manuscript stage and should be uploaded as a separate file during online submission labelled as File Type 'Abstract Figure'. Please ensure that you include the figure legend in the main article file. All Abstract Figures should be created using BioRender. Authors should use The Journal's premium BioRender account to export high-resolution images. Details on how to use and access the premium account are included as part of this email.

Done.

EDITOR COMMENTS

Reviewing Editor:

The current manuscript has been reviewed by two experts in the field. Both highlight the potential for this manuscript to enhance knowledge of exercise metabolism. There are however, some important comments that require careful consideration by the authors and appropriate amendments to the manuscript before it can be considered any further.

Authors: We sincerely appreciate the Editor's and Reviewers' comments. We have revised our manuscript following all of their recommendations. Please find below our itemized point-by-point response. Additions and corrections are marked (red font) in the revised manuscript.

REFEREE COMMENTS

Referee #1:

Thank you for the opportunity to review this paper on such a fascinating population. The study itself demonstrates that maximal fat oxidation occurs at near maximal exercise capacity in McArdle's patients, and the absolute rates of maximal fat oxidation that they exhibit are similar to well-trained disease-free individuals.

Below are my comments that the authors should consider in a revised manuscript.

Authors:
Comments much appreciated. Thank you.
Please find an itemized point-by-point response below.

Line 4 - remove "no"?
Done.

Line 106 - should be higher MFO 'rate'. This needs to be amended throughout.
Done (throughout the entire manuscript, including text as well as the relevant figure).

Line 151-206 - Was there any dietary control in the day(s) prior to the fatmax test? Or even information on habitual diet and whether this was similar between patients and controls. Unfortunately, diet was not controlled for in the day/s prior to testing, which is now acknowledged as a methodological limitation in the Discussion section since diet can affect FATmax. We hope that this limitation is attenuated, at least partly, by the fact that all participants attended the laboratory at the same time of the day (between 07.00 and 10.00 a.m.) after a long overnight fast (10 to 12 hours, with no between-group differences, as now mentioned in the results section). In addition, as also stated in the manuscript, participants were asked to refrain from strenuous physical activity and to avoid alcohol, tobacco, and caffeine consumption the day before assessments, in order to minimize additional potential confounders.

Line 218 - Why was the gastrocnemius muscle chosen? It would be useful to briefly justify the selection.

As now stated in the revised manuscript (methods section), '*The gastrocnemius muscle was chosen for the same reason as in previous research by our group with the McArdle mouse (Brull et al., 2015), that is, this is overall an 'intermediate' skeletal muscle in terms of the predominant metabolic phenotype as opposed to other muscles with a more characteristic oxidative/slow-twitch (e.g., soleus) or glycolytic/fast-twitch (e.g., extensor digitorum longus) profile, thereby providing a representation of the biological responses in the different main fiber types (I, IIA and IIB) that can be found in the adult mouse skeletal muscle.*

Line 299 - Report age in whole numbers in Table 1.
Done. (Of note, former Table 1 has now become Table 2).

Figure 3 - panel C, the -1 in watts.kg should be superscript.
Thanks for pointing out this typo. It has been corrected (of note, Fig. 3 has now become Fig 4).

Line 329 - what do you mean by steady-state? Resting? Can this be clarified please.
This term referred to protein levels under the conditions of our measurements (*i.e.*, rates of protein translation and degradation were assumed to be balanced). Anyway, the term 'steady-state' has now been removed as it is not strictly necessary at all.

Line 335 - How have you determined that there is 'voids full of glycogen granules'? The perfect solution here would be to stain using PAS or if you have access to a glycogen antibody (e.g. <https://pubmed.ncbi.nlm.nih.gov/24204959/>) and provide a representative image. This would add clarity especially for a non-expert, and the images would be very interesting for a reader who has a good understanding of microscopy. It could also be possible to provide a semi-quantitative assessment of glycogen content; this would be useful as to address the limitation of a lack of glycogen quantification.

All the parts of the manuscript related to the CD36 results obtained with immunofluorescence have now been removed as per the recommendation of the other Reviewer since this variable was not measured quantitatively. This way we believe the manuscript is now more focused and there is no room for speculation anymore. On the other hand, in the limitations section we have mentioned the fact that we did not quantify muscle glycogen (albeit we have previously done so in previous research with McArdle vs wild-type mice to ascertain massive depots in the former, as also mentioned in the revised Discussion section).

Line 337 - need to define what panel C in Figure 4 illustrates.
This panel has now been removed as per the recommendation of the other Reviewer (please see above response) because it referred to representative immunofluorescence staining images for CD36.

Line 391 - Suggest to re-word this. Only Orngreen et al. (2009) have measured plasma FFA concentrations to show that there was increased mobilization of fatty acids during exercise. This was not measured in the present study.
Reviewer 1 is quite right. We have reworded this sentence as follows (additions in boldface):
'*The increased mobilization of fatty acids during exercise reported by Ørngreen et al. (Ørngreen et al., 2009) together with the larger fat oxidation rates observed through indirect calorimetry in the present study suggest that patients with McArdle disease reach higher MFO rate values than their peers with preserved capacity for muscle glycogen utilization.*

Line 412-420 - Initially it is stated that the impaired oxidative capacity (and lower work capacity) in patients is related to the compromised rate of the TCA cycle. But then it is explained that carbohydrate feeding/infusion can overcome this intolerance. Presumably then, the TCA cycle isn't compromised per se in patients, since glucose oxidation would require this to be functional in order to improve work capacity. So is the TCA cycle truly compromised in patients or not?

Thank you for this comment. In McArdle patients, carbohydrate ingestion/infusion before exercise does not fully attenuate exercise intolerance in general and certainly does not fully compensate for oxidative impairment because.

Indeed, although the glycogenolytic flux in skeletal muscle fibers is blocked upstream the uptake of blood-borne glucose by muscle fibers (and subsequent metabolism of this substrate in these cells upon its conversion into glucose 6-phosphate), glycogen breakdown is still fully blocked anyway, which cannot be changed unless the deficient enzyme (myophosphorylase) is replaced (e.g., with gene therapy, and this is not expected to occur in the foreseeable future). Please see how we have rewritten this part:

'...Delaney et al. elegantly established a direct relationship between glycogenolysis and TCAI expansion during exercise, supporting the notion that impaired oxidative capacity in McArdle disease is related to a reduced TCAI availability, thereby inducing a metabolic bottleneck for β -oxidation-derived acetyl-CoA oxidation (Delaney et al., 2017). In this regard, it is well known that oral sucrose intake (Vissing & Haller, 2003b; Andersen et al., 2008) or intravenous glucose infusion (Haller & Lewis, 1991; Haller & Vissing, 2002) in patients with this condition increases the availability of blood glucose for working fibers, which is subsequently metabolized by skeletal muscle (i.e., bypassing the metabolic block in glycogenolysis), thereby attenuating exercise intolerance at submaximal workloads and partly improving peak work capacity. In any case, carbohydrate ingestion/infusion before exercise does not fully compensate for the inherited blockade in glycogenolysis upstream the uptake of blood glucose by muscle fibres and thus for the impairing effects of this block on the TCA rate (and therefore on oxidative metabolism): indeed, even after ingesting very high amounts of carbohydrates (75 g of fructose) 30 minutes before exercise, the VO_{2peak} of virtually all patients with McArdle disease (including active physically individuals) are still well below their age-/sex-matched normality value (57% lower on average) (Munguía-Izquierdo et al., 2015)'.

Line 439-440 - Can you expand on this adaptation from a mechanistic perspective? Perhaps there is greater CD36 located at the mitochondria as well as the plasma membrane in patients, in order to facilitate fatty acid entry into the mitochondria (<https://pubmed.ncbi.nlm.nih.gov/16670153/>).

Please see our above comment (i.e., we have now removed CD36 qualitative (i.e., immunofluorescence) data as per the recommendation of the other Reviewer and this part of the Discussion has now been removed accordingly.

Line 446 - Just clarify in this sentence that you are referring to patients rather than healthy individuals.

Done. We have changed 'this population' to 'patients with McArdle patients' to be precise.

Whilst I appreciate the effort the authors have gone to discuss the limitations, I believe there should also be a comment on the use of a single test, in light of data suggesting that there is poor day-to-day reliability in measures of fatmax (e.g. Chrzanowski-Smith et al., 2020, EJAP). How may have this affected the results of the study? Related to this, it would also be pertinent to comment on the lack of dietary control prior to the exercise testing.

Thanks for this comment. We have now mentioned in the limitations section the potential impact of the poor day-by-day reliability of MFO and FATmax measurements:

'In addition, although all participants were tested after an overnight fast, dietary intake was not controlled for in the day/s prior to the testing, which could be a confounding factor for substrate oxidation. It is also worth mentioning that the study participants were assessed only once; in this effect, determination of substrate oxidation rates by gas exchange measurements as we did here shows a large day-by-day variability (Chrzanowski-Smith et al., 2020), which might have confounded, at least partly, our results'.

On behalf of all coauthors, many thanks for this insightful review. It is much appreciated.

Referee #2:

In this study, the authors investigated the effect of exercise intensity on maximal fat oxidation rate in physically active McArdle patients and healthy controls. This was combined with measures of molecular markers of fat metabolism in skeletal muscle and white adipose tissue from McArdle and wild type mice. They find that McArdle patients achieve very high and maximal fat oxidation rates at near maximal exercise intensity, which suggest that in the condition of blocked glycogenolysis the maximal fat oxidation rate can be a limiting factor for exercise tolerance. The study is original and very well conducted and will impact the area of research by linking glycogen availability to very high fat oxidation rates even in non-athletes. Their conclusion is valid. I have two major comments and a few minor comments. The first major comment is about the reason for excluding 9 patients from the analyses. Although the curves of fat oxidation rate versus exercise intensity look different from the included curves and the mean values of PetCO₂ at VO₂ peak and the VE-VCO₂-1 at VO₂ peak are different, the reason for excluding each of the participants is missing. The second major comment is about data handling and interpretation of the findings from the samples collected from wild type and McArdle mice.

Authors: Comments much appreciated. Thank you.

Please find a point-by-point response below.

Major comments:

1. The manuscript can be improved by including more details on the reason for the exclusion of data from some McArdle patients (line 198 and 288). The authors should also consider moving the figures presented in the supplemental file to the main article text and present much more detail on how the potentially biased values were identified and if the PetCO₂ at VO₂ peak or the VE-VCO₂-1 at VO₂ peak could be used as a criterion for exclusion. In supplemental figure 3B and 3C there seems to be an overlap of the SD values between groups, which could indicate that some of the excluded McArdle patients had values close to the values of included patients and the controls. It would therefore be clearer if the individual values (PetCO₂ at VO₂ peak and the VE-VCO₂-1 at VO₂ peak) are shown in figure S3B and S3C with specific comments on why some patients (if any) with normal PetCO₂ at VO₂ peak and VE-VCO₂-1 at VO₂ peak values are excluded. These details are important for the understanding of why some data are excluded and the potential confounding role of hyperpnea.

Comment appreciated. We have followed all the recommendations by the Reviewer, with a detailed explanation of the reasons for excluding 9 patients due to excess hyperpnoea now added in the second paragraph of the Results section (under the subheading 'Study patients'), together with a new Figure 2 (i.e., former supplemental figure 3), which shows all individual

values of PetCO_2 and $\text{VE}\cdot\text{VCO}_2^{-1}$, respectively, at the last stage of the incremental step tests (i.e., FATmax protocol).

2. The collection of muscle samples from wild-type and McArdle mice is an important addition to the manuscript. However, given the relative low sample size and the high variability in data between muscles the findings remain rather inconclusive. Therefore, the manuscript could be improved by a presentation and discussion of data, which include information about how certain the findings are. This could be confidence intervals for the differences between groups and a discussion on whether the findings are physiological relevant and/or if the results are too uncertain to make any clear conclusion. The inclusion of immunofluorescence analyses of CD36 (Fig 4C) seems very preliminary (only qualitative observations are provided) and should therefore either be removed from the manuscript or the images should be analyzed quantitatively.

Comment appreciated. As we did for patients (and for the sake of consistency in the presentation of our results), we have now shown the magnitude of the effect (Hedge's g) for the two variables that showed significant differences between the two mouse groups (i.e., HADH in muscle tissue and pAmpkThr172 in white adipose tissue). On the other hand, we have now mentioned the limitation of the low sample size (together with the high variability) of the mouse results in the limitations section of the Discussion.

On the other hand, qualitative (i.e., immunofluorescence) analyses of CD36 have now been removed from the manuscript following the Reviewer's recommendation, so that the revised Discussion is much more focused, with much less speculation.

Minor comments:

Line 40: The wording of key point 5 seems wrong. Should it read "An animal model revealed no..."?

Yes indeed, corrected now. Thanks for catching this.

Line 58: It should be clearly stated that these data refer to data obtained from mice: "No between-group differences were found in molecular markers".
Done: *"No between-group differences were found in molecular markers in mice."*

Line 152: If possible, more detailed information on the overnight fast should be included. Within the current statement participants could be fasting for very different durations depending on the specific time of day for the testing and their last meal the evening before. Some may have been fasting for 12-14 hours while others for only 7-8 hours. Since fasting increases fat oxidation (PMID: 9931180) any difference in the fasting duration could affect the results. Therefore, the manuscript could be improved by more specific information on the fasting duration and/or discuss how this can have affected the results.

Done. Please see the following two additions in the revised manuscript:

. *'All participants attended the laboratory between 07.00 and 10.00 a.m. after an overnight fast of 12 to 14-hour duration'* (under the subheading 'Assessments').

. *'Overnight fasting lasted 13 ± 2 and 13 ± 1 hours in patients and controls, respectively ($p=0.767$ for the between-group difference)'* (second paragraph of the Results section).

Line 181: Correct wording "and/or"?

Thanks for catching this. We now have just used 'or' (which actually means/includes both 'and'

and 'or'). Same in other parts of the manuscript

Fig 4C is not mentioned in the legend.

Fig 4 has now become Fig 5 but, anyway, former panel C has been simply removed as it referred to immunofluorescence analyses.

On behalf of all coauthors, many thanks for this insightful review. It is much appreciated.

END OF COMMENTS

Dear Dr Ara,

Re: JP-RP-2022-283743R1 "Muscle glycogen unavailability and fat oxidation rate during exercise: Insights from McArdle disease" by Carlos Rodriguez-Lopez, Alfredo Santalla, Pedro L Valenzuela, Alberto Real-Martinez, Monica Villareal-Salazar, Irene Rodríguez-Gómez, Tomàs Pinós, Alejandro Lucia, and Ignacio Ara

I am pleased to tell you that your paper has been accepted for publication in The Journal of Physiology.

NEW POLICY: In order to improve the transparency of its peer review process, The Journal of Physiology publishes online as supporting information the peer review history of all articles accepted for publication. Readers will have access to decision letters, including all Editors' comments and referee reports, for each version of the manuscript and any author responses to peer review comments. Referees can decide whether or not they wish to be named on the peer review history document.

The last Word version of the paper submitted will be used by the Production Editors to prepare your proof. When this is ready you will receive an email containing a link to Wiley's Online Proofing System. The proof should be checked and corrected as quickly as possible.

Authors should note that it is too late at this point to offer corrections prior to proofing. The accepted version will be published online, ahead of the copy edited and typeset version being made available. Major corrections at proof stage, such as changes to figures, will be referred to the Reviewing Editor for approval before they can be incorporated. Only minor changes, such as to style and consistency, should be made a proof stage. Changes that need to be made after proof stage will usually require a formal correction notice.

All queries at proof stage should be sent to TJP@wiley.com.

Are you on Twitter? Once your paper is online, why not share your achievement with your followers. Please tag The Journal (@jphysiol) in any tweets and we will share your accepted paper with our 23,000+ followers!

Yours sincerely,

Michael C. Hogan
Senior Editor
The Journal of Physiology
<https://jp.msubmit.net>
<http://jp.physoc.org>
The Physiological Society
Hodgkin Huxley House
30 Farringdon Lane
London, EC1R 3AW
UK
<http://www.physoc.org>
<http://journals.physoc.org>

P.S. - You can help your research get the attention it deserves! Check out Wiley's free Promotion Guide for best-practice recommendations for promoting your work at www.wileyauthors.com/eeo/guide. And learn more about Wiley Editing Services which offers professional video, design, and writing services to create shareable video abstracts, infographics, conference posters, lay summaries, and research news stories for your research at www.wileyauthors.com/eeo/promotion.

*** IMPORTANT NOTICE ABOUT OPEN ACCESS ***

To assist authors whose funding agencies mandate public access to published research findings sooner than 12 months after publication The Journal of Physiology allows authors to pay an open access (OA) fee to have their papers made freely available immediately on publication.

You will receive an email from Wiley with details on how to register or log-in to Wiley Authors Services where you will be able to place an OnlineOpen order.

You can check if your funder or institution has a Wiley Open Access Account here: <https://authorservices.wiley.com/author-resources/Journal-Authors/licensing-and-open-access/open-access/author-compliance-tool.html>.

Your article will be made Open Access upon publication, or as soon as payment is received.

If you wish to put your paper on an OA website such as PMC or UKPMC or your institutional repository within 12 months of publication you must pay the open access fee, which covers the cost of publication.

OnlineOpen articles are deposited in PubMed Central (PMC) and PMC mirror sites. Authors of OnlineOpen articles are permitted to post the final, published PDF of their article on a website, institutional repository, or other free public server, immediately on publication.

Note to NIH-funded authors: The Journal of Physiology is published on PMC 12 months after publication, NIH-funded authors DO NOT NEED to pay to publish and DO NOT NEED to post their accepted papers on PMC.

EDITOR COMMENTS

Reviewing Editor:

Thank you for addressing the comments raised in review and congratulations on an insightful piece of work.

REFEREE COMMENTS

Referee #1:

Thank you for taking my feedback on board - I hope it was constructive. I enjoyed the opportunity to read the paper.

Referee #2:

Thank you for your response and revised manuscript. I have no further comments.

1st Confidential Review

21-Oct-2022